# Atomic scale disorder and reconstruction in bulk infinite-layer nickelates lacking superconductivity

Kejun Hu [1,4], Qing Li [2,4], Dongsheng Song [1,4] ✉, Yingze Jia[1], Zhiyao Liang[1], Shuai Wang[1], Haifeng Du [3], Hai-Hu Wen [2] ✉ & Binghui Ge [1] ✉

The recent discovery of superconductivity in infinite-layer nickelate films has sparked significant interest and expanded the realm of superconductors, in which the infinite-layer structure and proper chemical doping are both of the essence. Nonetheless, the reasons for the absence of superconductivity in bulk infinite-layer nickelates remain puzzling. Herein, we investigate atomic defects and electronic structures in bulk infinite-layer $Nd_{0.8}Sr_{0.2}NiO_2$ using scanning transmission electron microscopy. Our observations reveal the presence of three-dimensional (3D) block-like structural domains resulting from intersecting defect structures, disrupting the continuity within crystal grains, which could be a crucial factor in giving rise to the insulating character and inhibiting the emergence of superconductivity. Moreover, the infinite-layer structure, without complete topotactic reduction, retains interstitial oxygen atoms on the Nd atomic plane in bulk nickelates, possibly further aggravating the local distortions of $NiO_2$ planes and hindering the superconductivity. These findings shed light on the existence of structural and atomic defects in bulk nickelates and provide valuable insights into the influence of proper topotactic reduction and structural orders on superconductivity.

The emergence of superconductivity in nickelates represents a pivotal breakthrough in the exploration of superconducting materials and mechanisms[1], which is underscored by the notable similarity in the atomic configuration and electronic characteristics between the infinite $NiO_2$ layers in nickelates and the $CuO_2$ planes in cuprates[2–6]. The initial discovery occurred in $Nd_{0.8}Sr_{0.2}NiO_2$, achieved through the topotactic chemical reduction of the perovskite precursor $Nd_{0.8}Sr_{0.2}NiO_3$ using $CaH_2$[7–11]. Subsequently, other nickelates thin films with diverse compositions, such as $R_{1-x}A_xNiO_2$($R$ = La, Pr, Nd, and A = Sr, Ca, Eu), have demonstrated superconductivity under appropriate doping concentrations[12–19].

Two crucial factors contributing to the observed superconductivity are the infinite layer structure and proper hole doping achieved through the alkaline earth substitution[20]. The infinite-layer structure is typically derived from the precursor perovskite phase. However, synthesizing a pure perovskite phase, i.e., $Nd_{0.8}Sr_{0.2}NiO_3$, remains challenging due to structural instability[20]. This instability is primarily attributed to the high tensile strain in precursor films grown on substrates[19–21] and the instability of the high-valence state $Ni$[18,22]. Consequently, the Ruddlesden-Popper phase ($R_{n+1}Ni_nO_{3n+1}$, $R$ is rare-earth) is often observed in nickelates thin films, hindering the manifestation of superconductivity[23,24]. Recent developments even

[1]Information Materials and Intelligent Sensing Laboratory of Anhui Province, Anhui Key Laboratory of Magnetic Functional Materials and Devices, Institutes of Physical Science and Information Technology, Anhui University, Hefei, China. [2]Center for Superconducting Physics and Materials, National Laboratory of Solid State Microstructures and Department of Physics, Nanjing University, Nanjing, China. [3]Anhui Key Laboratory of Condensed Matter Physics at Extreme Conditions, High Magnetic Field Laboratory, HFIPS, Anhui, Chinese Academy of Sciences, Hefei, China. [4]These authors contributed equally: Kejun Hu, Qing Li, Dongsheng Song. ✉e-mail: dsong@ahu.edu.cn; hhwen@nju.edu.cn; bhge@ahu.edu.cn

show that intentionally grown layered RP phase $La_3Ni_2O_7$ can exhibit superconductivity with a $T_c$ of 80 K under pressure[25]. However, this superconducting mechanism differs from that of infinite-layer nickelates by stabilizing intermediate nickel valences[25,26]. These orthorhombic RP phases ($R_{n+1}Ni_nO_{3n+1}$) typically transform into the tetragonal T′-type phase ($R_{n+1}Ni_nO_{2n+2}$) through oxygen deintercalation and structural rearrangement after topotactic reduction[10,27]. The T′-type phase possesses quasi-two-dimensional $NiO_2$ planes in square-planar coordination with rare-earth fluorite blocking slabs interleaved in every $n$ nickel layer[7,28–30]. These intricate phenomena manifest the strong correlation between the local atomic structure and superconductivity in nickelates. For the hole doping mechanism, the electronic structure measurements have provided insights within the Mott-Hubbard regime in the parent phase $NdNiO_2$[6]. The doping evolution has been convincingly demonstrated through locally resolved electron energy-loss spectroscopy (EELS)[31]. The interfacial reconstruction has been revealed to stabilize the external crystal structure and alleviate polar discontinuity in $Nd_{0.8}Sr_{0.2}NiO_2$ thin films[32–35]. Despite significant strides in addressing these key issues and advancing our understanding of superconductivity in nickelates thin films, the reasons for the absent superconductivity in bulk nickelates remain elusive[36–38].

Herein, using integrated differential phase contrast (iDPC) and electron energy-loss spectra (EELS) combined with state-of-the-art aberration-corrected scanning transmission electron microscopy (STEM), the microstructure and atomic structure of polycrystalline $Nd_{0.8}Sr_{0.2}NiO_2$ are resolved to unveil the origin of absent superconductivity in bulk nickelate. A large number of structural defects and atomic distortions were found therein to contribute to a three-dimensional (3D) block-like structure, causing the breakdown of connections within the grains. The impurity fluorite-layers are unexpectedly discovered across the T′-type phase boundaries, attributing to the impurity of reducing agent $CaH_2$. Furthermore, iDPC-STEM imaging reveals the residual O atom at the apical of $NiO_6$ octahedra after topotactic reduction, introducing additional distortions in infinite layers. The effects of structural defects on the transport property and superconductivity of polycrystalline $Nd_{0.8}Sr_{0.2}NiO_2$ are discussed.

## Results

### Structural defects in bulk $Nd_{0.8}Sr_{0.2}NiO_2$

The precursor $Nd_{0.8}Sr_{0.2}NiO_3$ was synthesized via a two-step high-temperature and high-pressure solid-state synthesis method, followed by topotactic reduction using $CaH_2$ (Aladdin, 98.5%, Mg <1%) at low temperature to obtain the polycrystalline bulk $Nd_{0.8}Sr_{0.2}NiO_2$. While the parent phase $Nd_{0.8}Sr_{0.2}NiO_3$ has an orthorhombic structure ($Pbnm$) with lattice parameters of $a_o = 5.39$ Å, $b_o = 5.38$ Å, and $c_o = 7.61$ Å, here we use the pseudocubic unit cell to define the orientation with lattice parameters of $a_{pc} = b_{pc} = 3.808$ Å, $c_{pc} = 3.805$ Å. The orientation relationship between the pseudocubic indices and the orthorhombic indices of the $Nd_{0.8}Sr_{0.2}NiO_3$ are $[010]_{pc} // [110]_o$, $[100]_{pc} // [1\bar{1}0]_o$, $[001]_{pc} // [001]_o$. The infinite layer $Nd_{0.8}Sr_{0.2}NiO_2$ has a tetragonal structure with $a_t = b_t = 3.92$ Å, $c_t = 3.34$ Å. The atomic models before and after the topotactic transformation are illustrated in Fig. 1a. The energy dispersive X-ray spectra (EDS) mappings at low magnification show the uniform elemental distributions of Nd, Ni, O, and doped Sr in the polycrystalline in Fig. 1b. The presence of grain boundaries and voids in bulk nickelate, as shown in Fig. 1b, can indeed elevate resistance and make the material more insulating. As discussed in a previous study[36], applying high pressure can mitigate the impact of grain boundaries on the transport properties through increased density and reduced voids. However, despite the weakening of insulating behavior under high-pressure conditions, superconducting properties remained elusive. This suggests that the presence of grain boundaries and voids is not the primary detrimental effect on superconductivity. Figure 1c illustrates the temperature dependence of resistivity for bulk

$Nd_{0.8}Sr_{0.2}NiO_2$ under ambient pressure. Notably, it exhibits strong insulating behavior over the temperature range of 2 to 300 K, indicating the absence of superconductivity in bulk $Nd_{0.8}Sr_{0.2}NiO_2$.

The high-angle annular dark-field scanning transmission electron microscopy (HAADF-STEM) image magnified in Fig. 1d reveals the presence of numerous stripes within the $Nd_{0.8}Sr_{0.2}NiO_2$ grains. Further examination of the grain interiors provides clear stripe contrast, as shown in Fig. 1e, f. Some of these stripes align parallel to the [010] direction (highlighted by a yellow box in Fig. 1d), with the elongated diffraction spots extending along the [001] direction in the inset of Fig. 1e. For regions displaying a vertically staggered distribution (marked by a red box in Fig. 1d and Fig. 1f), the bi-directional splitting of the diffraction spots indicates that the area contains two domains with orthogonally oriented $c$-axis, as marked by the lattice vectors in the inset of Fig. 1f. The enlarged images of Selected Area Electron Diffraction (SAED) are presented in Supplementary Fig. S1, where two sets of lattice vectors are indexed. In addition, scanning diffraction measurements based on the four-dimensional STEM (4D-STEM) technique, utilizing a hybrid pixelated detector, were applied to the interleaved region. The typical diffraction patterns extracted from the two orthogonal regions provide evidence of domains with orthogonal orientation, as further supported by the virtual 4D-STEM imaging (Supplementary Fig. S2) and atomic resolution images of sub-regions within the interleaved area (Supplementary Fig. S3). To gain insight into the grain orientation at the mesoscale, the crystalline orientation was determined using Transmission Kikuchi Diffraction (TKD) for the TEM sample ("Methods" and Supplementary Fig. S4). The results showed a random orientation without any discernible textures. Consequently, the polycrystalline nature, in conjunction with the abundance of grain boundaries, is expected to increase resistance and make the material more insulating, in contrast to single crystal thin films.

Subsequently, a detailed analysis of the stripe-like structure is conducted. Atomic resolution HAADF-STEM and iDPC-STEM with the capability of imaging heavy and light elements simultaneously are used to identify these phases as the T′-type ($R_{n+1}Ni_nO_{2n+2}$, $R$ is rare-earth) structure, which is classified by the number of $NiO_2$ layers $n$. As illustrated in Fig. 2a, taking $n = 3$ ($R_4Ni_3O_8$) as an example, the apical oxygen of $NiO_6$ octahedra in the parent RP phase ($R_4Ni_3O_{10}$) is deintercalated after the topotactic reduction. This conversion changes the $RO$ planes by sole element $R$, accompanied by the transformation of $NiO_6$ octahedra into $NiO_2$ square-planar. Furthermore, the rock salt layers of the RP phase undergo a transformation into the fluorite structure after topotactic reduction, involving the rearrangement of oxygen atoms between two $R$ layers, as illustrated by the red spheres in Fig. 2a. Figure 2b displays a typical iDPC-STEM image of the T′-type phase, highlighting the formation of fluorite slabs and infinite $NiO_2$ layers. The rearranged oxygen layer is indicated by orange arrow. The HAADF-STEM image in Fig. 2c reveals that the T′-type phases with different layers number $n$ are alternatively arranged along the $c$ axis separated by the fluorite layers. The intensity profile across the interface is plotted in Fig. 2d. In addition, the intensity of $R$ atom columns near the fluorite layers is relatively weaker than that in infinite layers, as indicated by the purple stars in Fig. 2d, implying the changed chemical composition according to the $Z$ contrast character of HAADF-STEM imaging. In consideration of the co-occupying Sr and Nd atoms at $R$ sites, the decreased intensity reveals a higher ratio of Sr/Nd localized at the fluorite layers. The decrease in lattice spacing within the fluorite layers is clearly demonstrated by the bond lengths depicted in Fig. 2e. This finding is consistent with the outcomes of the geometrical phase analysis (GPA) presented in Fig. 2f, where the compressive strain $\varepsilon_{yy}$ attains a negative peak at the fluorite layers.

### Chemical impurities in bulk $Nd_{0.8}Sr_{0.2}NiO_2$

Furthermore, high-magnification STEM-EDS measurements were carried out to analyze the element distributions in Fig. 3a, b. For the

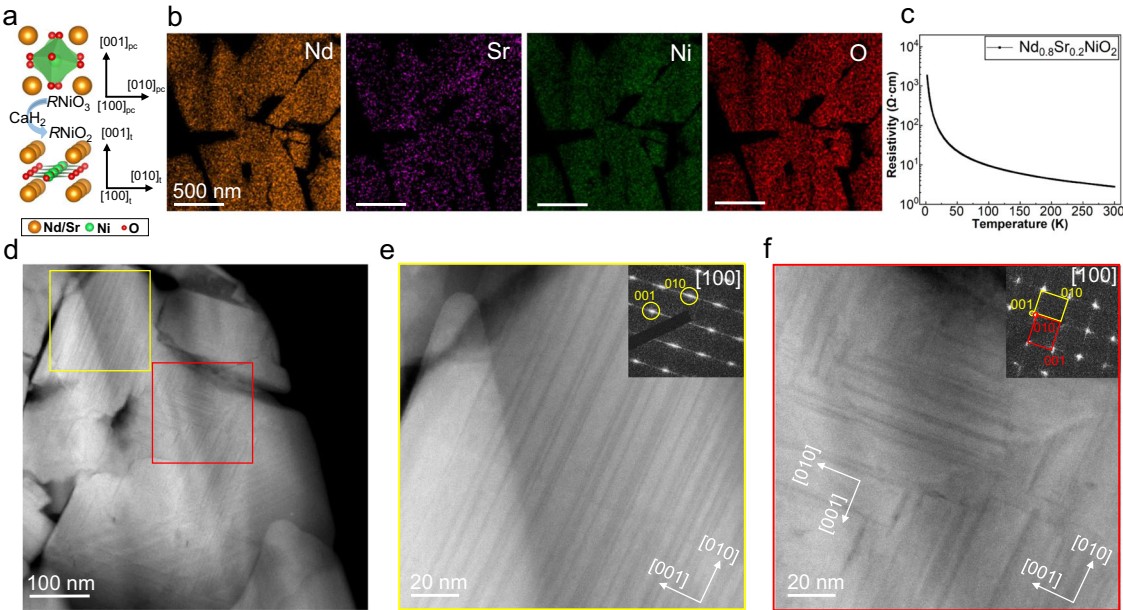

**Fig. 1 | Polycrystalline Nd$_{0.8}$Sr$_{0.2}$NiO$_2$ imaged by HAADF-STEM. a** Atomic structure model with direction vectors of perovskite $R$NiO$_3$ and infinite layer $R$NiO$_2$ ($R$: Nd/Sr). **b** EDS mappings of Nd$_{0.8}$Sr$_{0.2}$NiO$_2$ at low magnification. **c** Temperature dependence of resistivity for bulk Nd$_{0.8}$Sr$_{0.2}$NiO$_2$ in the range of 2–300 K. **d** High-resolution HAADF-STEM image of Nd$_{0.8}$Sr$_{0.2}$NiO$_2$. **e, f** The magnified images of marked regions in (**d**), with the corresponding SAED pattern in the insets.

high-magnification EDS mapping in Fig. 3a, a lower filter setting is utilized, as described in the "Methods" section. The unfiltered data for Fig. 3a was also shown in Supplementary Fig. S6. The deficiency of Ni at the fluorite layers can be easily understood according to the atom model in Fig. 2a. Although the Nd seems to have a relatively uniform distribution, Sr is enriched at the fluorite layers, consistent with the increased ratio of Sr/Nd and thus the decreased contrast of HAADF-STEM image in Figs. 2c and 3a. The doped Sr plays a vital role in the multi-band properties and the minimum energy of infinite-layer nickelate, and the hole carriers provided by Sr are necessary for the superconductivity[12,39]. Moreover, the element F is unexpectedly observed at the fluorite layers, accompanied by a deficient oxygen distribution in Fig. 3b. It is noteworthy that the element F is not involved during the whole topotactic reduction process. We suppose that it might come from the impure CaH$_2$ as the precursor of CaF$_2$ could be introduced even with a small amount. It is further confirmed by the X-ray photoelectron spectra (XPS) measurements of CaH$_2$ in Fig. S5. Therefore, the chemical compositions of ingredients and target samples should be carefully examined before and after the topotactic reduction, as the unexpected impurities might play a significant role in superconductivity, such as the impurity H as discovered before[40]. The replacement of O by F atoms could change the valence state of Ni due to the strong electronegativity of F. Meanwhile, the fluorite layer might decouple the NiO$_2$ layers and suppress the hybridization between the $c$-axis as discussed in the case of La$_2$NiO$_3$F[41–43].

Moreover, atomic resolution EELS elemental mappings show similar distributions to the EDS results in Fig. S6. The EELS near-edge fine structures of O $K$, F $K$, Ni $L$, and Nd $M$ edges were measured both for the infinite-layer phase and T'-type phase, as depicted in Fig. 3c. To improve the signal-to-noise ratio, the EELS signals in Fig. 3c are integrated over the regions as marked in Supplementary Fig. S6b. The O $K$ edge shows almost no changes between the two phases. The distinct $K$ edge of F appears at around 685 eV for fluorite layers, further evidencing the existence of impurity F. Particularly, F only replaces oxygen at the fluorite layers and it is not discovered in the infinite NiO$_2$ layers. Although the fine structures of the Nd $M$ edge show no difference, the prominent increased intensity of the $L_3$ edge for Ni is observed for T'-type phase. The larger ratio of $L_3/L_2$ is generally

attributed to the valence changes, i.e., a higher valence of Ni in the T'-type phase. However, the peak energies of $L_3$ and $L_2$ edges remained nearly unchanged between these two phases. In this complex system, factors such as Sr doping, variations in oxygen content, and changes in the coordination environment can alter the shape, intensity, and energy position of the $L_{3,2}$ edges for Ni[31]. Therefore, we believe the different coordination environments of Ni between the T'-type phase and the infinite layer might contribute to this observed phenomenon.

The segregation of Sr near fluorite layers may result in lower doping within the infinite-layer regions, potentially responsible for the absence of superconductivity. To explore this possibility, we compare the EELS results of the infinite-layer phases with those in superconducting infinite-layer nickelates thin films, which are nicely and systematically measured in a prior study[31]. As shown in Supplementary Fig. S7, the pre-peak of the O $K$ edge, indicative of the doping level[31], is weaker than that in the superconducting thin films. This observation aligns with the underdoped concentration of Sr in the infinite-layer phase due to Sr segregation near fluorite layers. The shift of Ni $L_{3,2}$ edges towards lower energy in Supplementary Fig. S7b is also consistent with the lower Sr doping, despite the broadening of the peaks, which might be attributed to the relatively lower EELS energy resolution in our experiments (~1 eV). However, given that the Ni $L_3$ peak energy is expected to increase with higher Sr concentrations and is also sensitive to the oxygen content[31], the decrease in the Ni $L_3$ peak energy observed here could be attributed not only to the lower doping level of Sr in the infinite layers, but also to variations in oxygen content, due to the presence of residual apical oxygen atoms. As the infinite-layer structure, the proper concentration of doped Sr and the oxygen content are essential to superconductivity, the presence of various T'-type phases with such imperfections would probably lead to the insulating and non-superconducting characters in nickelates[20,36,44].

### 3D block-like configurations

In addition to the T'-type phases arranged vertically to the $c$-axis (parallel stripes), the stripes parallel to the $c$-axis (vertical stripes) are also formed as shown in Fig. 4a, highly intersected with the parallel ones. Figure 4b shows an enlarged iDPC image of the vertical stripes and the atomic structure model is displayed on the upper side. In

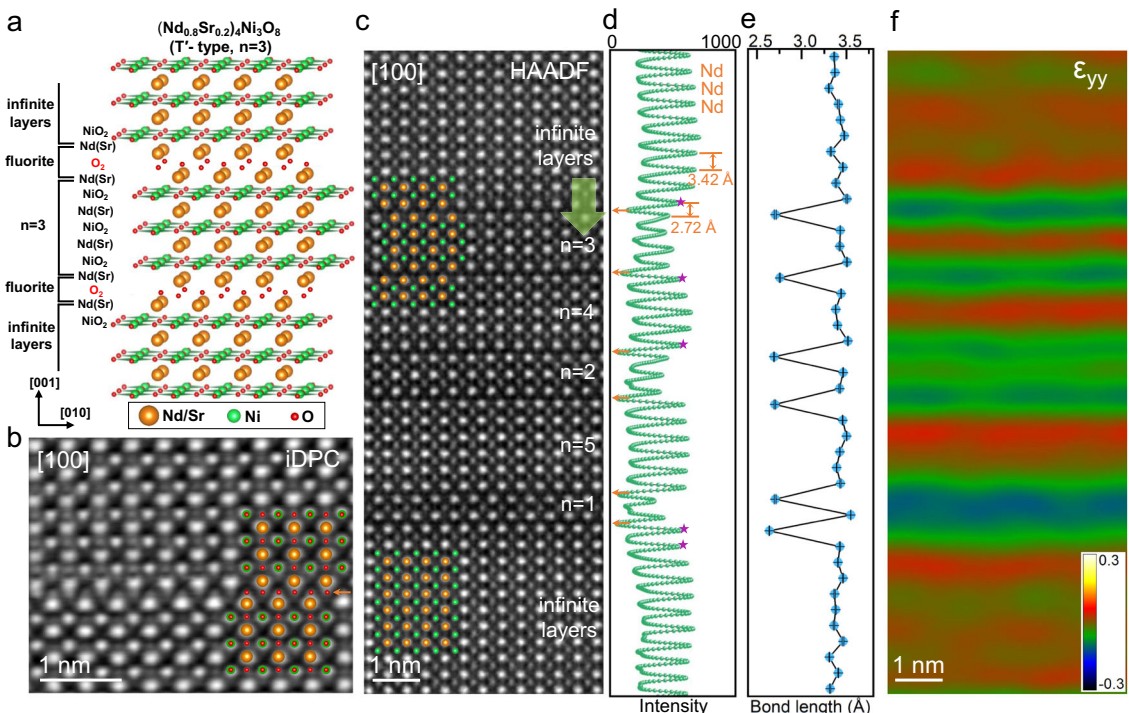

**Fig. 2 | Atomic structure of the stripe domains in $Nd_{0.8}Sr_{0.2}NiO_2$. a** Atom model of layered T′-type $(Nd_{0.8}Sr_{0.2})_4Ni_3O_8$ ($n = 3$). **b** Atomic-resolved iDPC-STEM image of the T′-type phase along [100] direction with the light elements in infinite and fluorite layers clearly resolved. The atomic model is overlaid to guide the eyes. **c** Atomic-resolved HAADF-STEM image of the stripes under [100] orientation, and the atomic models without oxygen are overlaid accordingly. To differentiate between T′-type phase and infinite layer structures, the latter are simply defined here as $n > 10$ layers. **d** The intensity profile across adjacent Nd atomic columns is extracted along the direction indicated by the green arrow in (**c**). The orange arrows are used to mark the fluorite layers. **e** Averaged bond lengths between $R$-site atoms along [001] direction. **f** Strain $\varepsilon_{yy}$ along the [001] calculated by GPA for (**c**).

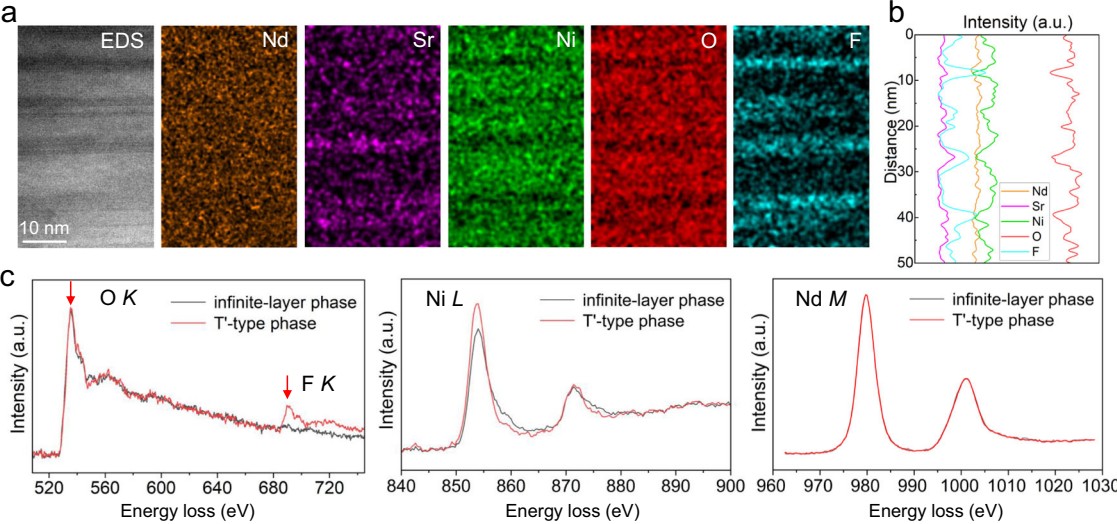

**Fig. 3 | EDS and EELS measurements of $Nd_{0.8}Sr_{0.2}NiO_2$. a** EDS elemental mappings of the stripe structure in $Nd_{0.8}Sr_{0.2}NiO_2$. **b** The intensity profile of elemental distributions in (**a**). **c** EELS O $K$, Ni $L$, and Nd $M$ of infinite-layer phase (black) and T′-type phase (red).

contrast to the T′-type phases with oxygen/fluorine layers, the vertical stripes are formed by a 1/2[101] shift of the infinite-layer phase (left) with respect to the other one (right), similar to the rock salt layer in the RP phase. The vertical and parallel stripes are distinctly discernible through GPA in Fig. 4c, d. To address potential artifacts associated with GPA[45], we simultaneously conducted measurements of bond lengths, as previously undertaken[46–48]. The variations in lattice spacing within these stripes are also clearly resolved, as illustrated in Fig. S8. These two types of defects are coherent with the infinite-layer phase, finally forming a 3D block-like configuration in the bulk nickelates.

To clarify the 3D block-like configurations clearly, the joint region is shown in Fig. 4e with the sample oriented along [001] direction. The boundaries are indicated by the green dotted lines. The relatively uniform contrast at each atomic column for Area 2 indicates two overlapping regions along [001] direction with a translation vector of 1/2[110], in contrast to that in Areas 1, 3, and 4. Based on the structural

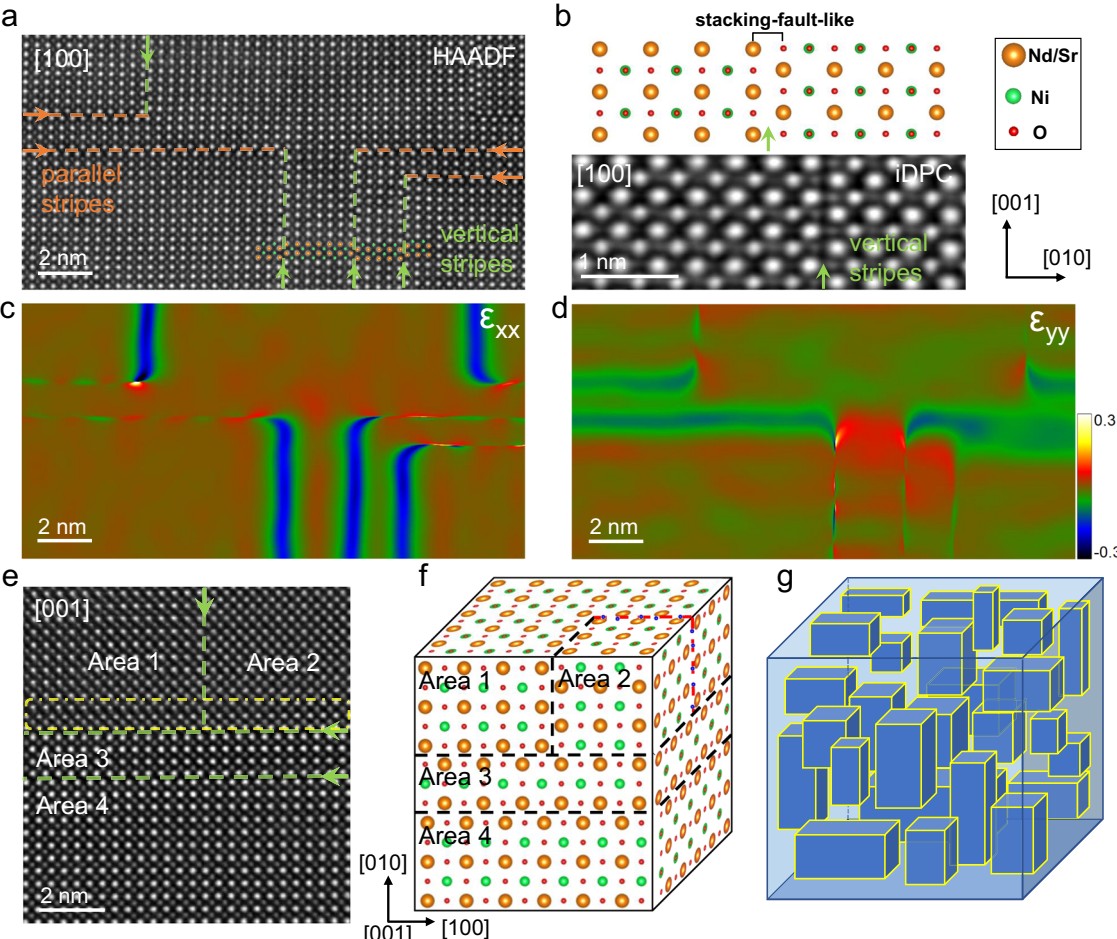

**Fig. 4 | 3D block-like configurations in Nd$_{0.8}$Sr$_{0.2}$NiO$_2$ grains. a** Atomic resolution HAADF-STEM images of the parallel and vertical stripes along the [100] zone axis. The orange and green arrows point to the interface of parallel stripes and vertical stripes, respectively. **b** Atom model of the vertical stripe structure. The enlarged iDPC-STEM image of the vertical stripe is displayed below. **c**, **d** The ε$_{xx}$ and ε$_{yy}$ strain maps are calculated by GPA for (**a**). **e** HAADF-STEM image of the regions containing several kinds of stripe structures along [001] zone axis, which is indicated by the green dotted line. In addition, the slight change of atomic column contrast in the yellow dotted box may be caused by the inclined wedge interface. **f** Schematic diagram of the 3D atomic model of the central joint area in (**e**), in which only 1 cell is drawn in the direction [010] in Area 3 due to space. The black dashed line represents the stacking-fault-like plane, and the red dashed line represents the fluorite layer of T′-type phases. **g** Schematic diagram of a 3D block-like model composed of randomly distributed stripe domains in the block.

analysis along [001] direction above, the 3D atom model is built in Fig. 4f. When viewed along [001] direction, the intersecting domain structures are consistent with the observations of the central region in Fig. 4e. We also compared the EELS fine structure with that of the infinite-layer phase in Fig. S9, no obvious difference is observed for O $K$, Ni $L$, and Nd $M$ edges, further confirming that the regions are still the infinite-layer phases. The polycrystalline Nd$_{0.8}$Sr$_{0.2}$NiO$_2$ are not only segmented by the grain boundaries but also disrupted by the stripe domains as schematically shown in Fig. 4g. These stripe structures are presented with the modified chemical composition and electronic structure within the crystal grains as discussed in Fig. 3, which could be the possible origin of insulating behavior and absent super-conductivity in bulk nickelates.

**Distortions in infinite layers**

At last, the local lattice distortions of the infinite-layer phase are analyzed as there is always a strong correlation between different degrees of freedom in perovskite oxides[49]. The soft chemical topotactic reduction reduces the perovskite $R$NiO$_3$ to the infinite layer $R$NiO$_2$ by removing the apical oxygen atoms in the distorted NiO$_6$ octahedron and thus generates a two-dimensional NiO$_2$ plane with square coordination, as schematically shown in Fig. 5a. iDPC-STEM technique was used to visualize the oxygen[50] in the infinite layers in Fig. 5b. Remarkably, there are still randomly distributed residual apical oxygen atoms in Nd/Sr planes after topotactic reduction, as indicated by the orange arrows in Fig. 5c–e. To exclude the possible imaging artifacts, iDPC-STEM simulations are conducted based on the perfect atomic model of the infinite-layer in Fig. S10. No extra contrast is observed at the positions of apical oxygen atoms. Therefore, it is suggested that a few residual apical oxygen atoms are still present in the bulk, which could be caused by the insufficient topotactic reduction due to the large volume of the bulk sample, unlike the thin films only with a thickness of tens of nanometers. Besides, the strain effect in thin films could degenerate the equivalent occupancy of oxygen atoms in NiO$_6$ octahedron and facilitate the removal of apical oxygen more easily, in contrast to the similar oxygen coordination environment in the bulk.

The presence of residual apical oxygen would generate a large interlayer coupling between intralayer Ni cations by altering the electron energy levels[25] and impact the flatness of the infinite NiO$_2$ planes by introducing local structural distortions, which could serve a crucial role in superconductivity. Hence, the detailed atom position analysis was performed to evaluate the local distortions in Fig. 5f, including the lattice parameter of $c/a$, the O–Ni–O bond angle, and the deviations of Ni atoms from the center of four-nearest Nd/Sr. The $c/a$ ratio exhibits a

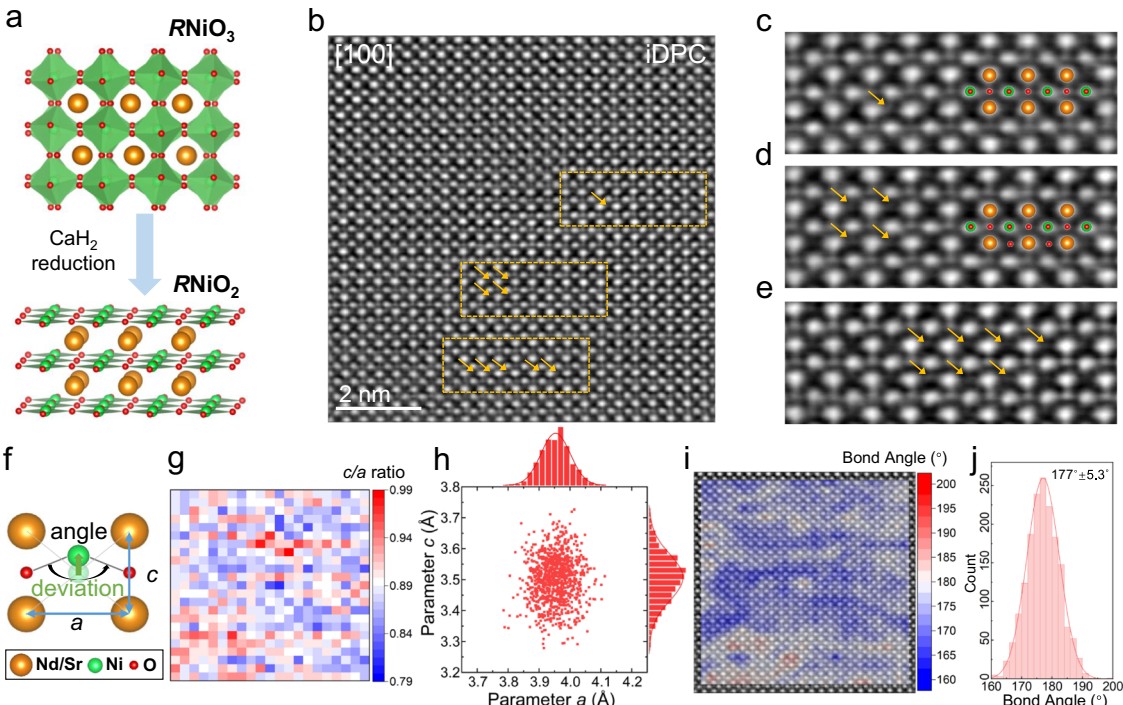

**Fig. 5 | Atomic iDPC-STEM imaging of infinite-layer phase. a** Structure transition during the topotactic reduction. **b** iDPC-STEM image of infinite-layer phase at [100] orientation. **c**–**e** Enlarged iDPC-STEM images extracted from the orange dotted line box in (**b**), where the orange arrow marks the residual oxygen atom in the Nd/Sr atomic plane. **f** Atom model of the distorted $NiO_2$ layers projected in [100] direction. The bond angle and the deviation of Ni atom are marked as well. **g** The ratio of lattice parameter $c/a$ according to (**b**). **h** Statistical data of lattice constants $a$ and $c$. **i** The calculated bond angle contour maps are plotted over the iDPC-STEM image. **j** The histogram displays the statistical data for bond angles measured across multiple regions, with a fitted value of $177.0° \pm 5.3°$.

wide fluctuation ranging from 0.81 to 0.99 in Fig. 5g, indicating the nonuniform structural distortions. The values $c/a$ measured on a large number of iDPC-STEM images are statistically plotted in Fig. 5h, leading to the average value of approximately $0.886 \pm 0.021$, comparable to the value of 0.86 in thin films at the same concentration of Sr doping[1].

The contour map of the O–Ni–O bond angle is displayed in Fig. 5i, where it is noticeably deviated from 180°, potentially impacting the flatness of the $NiO_2$ layers. To ensure the reproducibility and reliability of our measurements, we have calculated the O–Ni–O bond angles across various regions in multiple experiments, as detailed in Supplementary Fig. S11. The average bond angle is determined to be $177.0° \pm 5.3°$, as shown in Fig. 5j, with a broadened distribution. This finding highlights the presence of complex distortions in the Ni–O planes of polycrystalline $Nd_{0.8}Sr_{0.2}NiO_2$ after chemical reduction. These disorders within the $NiO_2$ layers are consistent with previous X-ray analysis of bulk polycrystalline samples, which have shown broad diffraction peaks[36].

## Discussion

The primary condition for the manifestation of superconductivity in nickelates is their metallic character, which is not observed in the bulk sample displaying insulating behavior. This may be induced by two main factors. Firstly, numerous stripe phases, such as the T′-type phase ($R_{n+1}Ni_nO_{2n+2}$), are present within the grains exhibiting insulating behavior[30]. These phases originate from the parent RP phase of the $Nd_{0.8}Sr_{0.2}NiO_3$ compound before undergoing topotactic reduction, as shown in Supplementary Fig. S12 and S13. Then, these RP phases were transformed into the T′-type phase during the subsequent topotactic reduction process. Although the $Nd_6Ni_5O_{12}$ compound with a layer number of $n = 5$ exhibits superconductivity through self-doping[30], the layer number varies from place to place in the grain of bulk samples.

Secondly, proper doping is essential for conferring metallic character and, consequently, inducing superconductivity. As discussed previously[12–14,31], the infinite-layer phase with a low hole doping level does not display superconductivity. Hence, despite the existence of a portion of the infinite-layer phase in the bulk sample with a certain amount of Sr, the doping level may be lower compared to the threshold required for the superconducting phase. This can be corroborated by the observation of the segregation of Sr across the interface of the T′-type phase, resulting in a low Sr content in the infinite layer, as shown in Fig. 3. The lower concentration of Sr doping is also revealed by the reduced intensity of pre-peak of the O $K$ edge in Supplementary Fig. S7, consistent with previous results[31]. Consequently, achieving superconductivity in bulk nickelates necessitates the preparation of a high-quality $Nd_{0.8}Sr_{0.2}NiO_3$ phase as the initial step. In cases where the RP phase cannot be completely eliminated, a higher concentration of Sr doping might be necessary to prevent underdoping caused by Sr segregation in the reduced RP phase (i.e., T′-type phase).

It is important to note that many thin films, i.e., nickelates[17,20] and $Sr_2RuO_4$[51], exhibit superconductivity even in the presence of a high density of defects similar to those found in bulk materials. However, these defects, such as non-superconducting layers or inclusions, are usually confined to the upper regions of the thin films[20]. In contrast, the superconducting layer with the infinite layer structure and proper hole doping near the substrate tend to be well-preserved throughout the sample or within a large area. Consequently, the signature of superconductivity is evident in electrical measurements. In the case of bulk materials, the infinite layer phases are indeed present, but the continuity of the $NiO_2$ planes within the grains is disrupted by numerous and randomly distributed stack faults and interleaved structures. Furthermore, these striped infinite-layer phases even undergo orientation flips within the grain, as illustrated in

Supplementary Fig. S14. Establishing a superconducting channel becomes challenging under these conditions. Therefore, eliminating or reducing these defects is necessary for achieving superconductivity in the bulk case.

At last, the incomplete removal of apical oxygen in $Nd_{0.8}Sr_{0.2}NiO_2$, as depicted in Fig. 5, introduces external structural distortion that impacts the flatness of the $NiO_2$ plane. This makes the $NiO_2$ planes buckle, leading to enhanced scattering of electrons and inhibiting the pairing[52–55]. Thus, this incomplete removal is considered to be the alternative possibility for the absence of superconductivity in the present bulk samples. The presence of residual oxygen atoms in the Nd/Sr plane disrupts the long-range order of the infinite-layer structure, leading to suppressed superconductivity characterized by a low transition temperature, as discussed in a prior study on thin films[56]. Moreover, the insufficient topotactic reduction observed in the bulk sample is likely attributed to its large volume compared to thin films, which typically have a thickness of tens of nanometers. As a result, it is crucial to engineer the process of topotactic reduction to optimize the infinite layer and ultimately induce superconducting.

In summary, the microstructure and atomic structure of polycrystalline nickelates were thoroughly analyzed by scanning transmission electron microscopy to reveal the origin of absent superconductivity in its bulk form. A large number of T′-type phases and stacking-fault-like defects are found to form the 3D block-like configurations, disrupting the continuity within crystal grains. The impurity F is unexpectedly discovered to replace the oxygen atoms in the fluorite layers. The fluorite layers in the T′-type phases are found to be Sr-rich and F-rich, leading to the change of valence state of Ni accordingly. Moreover, the residual oxygen atoms were observed in the Nd/Sr plane due to the incomplete topotactic reduction, which could distort the local structure and hinder the flatness of the $NiO_2$ plane. These structural defects and local distortions in infinite layers could be responsible for the insulating and non-superconducting characters in bulk nickelates. Our findings highlight the importance of proper topotactic reduction and structural order to superconducting properties and suggest the possible origin of absent superconductivity in nickelates.

## Methods

### Sample synthesis

To synthesize polycrystalline samples of $Nd_{0.8}Sr_{0.2}NiO_2$, the following procedure was employed[36]. Initially, polycrystalline precursor $Nd_{0.8}Sr_{0.2}NiO_3$ were prepared through a two-step solid-state reaction under high pressure and high-temperature conditions with $KClO_4$ as an excess oxygen source. The resulting $Nd_{0.8}Sr_{0.2}NiO_3$ samples were washed with distilled water. Subsequently, samples of $Nd_{0.8}Sr_{0.2}NiO_2$ were obtained via a topochemical reduction process from $Nd_{0.8}Sr_{0.2}NiO_3$ using $CaH_2$ as a reducing agent. Three molar additions of $CaH_2$ and one molar of $Nd_{0.8}Sr_{0.2}NiO_3$ were sealed in an evacuated quartz tube and subjected to heating at 280 °C for 20 hours. The resulting pellets were then crushed and washed with saturated $NH_4Cl$ in anhydrous ethanol to remove any residual $CaH_2$ and reacted byproduct CaO. Finally, the samples were dried in an evacuated oven to obtain the target sample of the infinite-layer phase in powder form. For resistivity measurements, the powders were pressed into a pellet and heat treated at 180 °C for 10 h together with a separately pellet of $CaH_2$.

### Preparation of transmission electron microscopy sample

The cross-sectional TEM lamellas of $Nd_{0.8}Sr_{0.2}NiO_2$ was thinned to electron beam transparency at 30 kV by using Carl Zeiss Crossbeam 550 L FIB-SEM, followed by the removal of the surface amorphous layer at 2 kV. In addition, to exclude potential structural transition or detects induced by the Ga ion beam during TEM sample preparation, we chose an alternative approach by grinding the sample, dispersing it into

alcohol, and subsequently transferring it to the TEM grid. Note that we abstained from using the polishing method due to the fragility of the reduced sample. As depicted in Supplementary Fig. S15, the observed stripe structures and defects remain consistent, providing additional evidence that rules out structural transitions or defects induced by FIB processing.

### Structural characterization

The HAADF-STEM, iDPC-STEM, and STEM-EELS experiments were conducted using a Thermo Fischer Scientific Titan Themis Z microscope operating at 300 kV, equipped with a probe corrector, a four segment DF4 detector, a super EDS detector and a Gatan GIF Continuum dual-EELS system. The convergence semi-angle for imaging is 25 mrad, and the collection semi-angle is 50–200 mrad for the HAADF imaging and 8–42 mrad for iDPC. The iDPC images were reconstructed from the four segment images, employing a high-pass filter to diminish low-frequency information. Both data acquisition and processing were carried out using the commercial Velox software. Regarding the EDS mappings in Fig. 1 and Supplemental Fig. S12 at low magnification, we have utilized the unfiltered data. For the high magnification EDS mappings in Fig. 3, we have utilized a post-filter setting (average pixel size: 7) to smooth these images. The optimal defocus value for HAADF imaging is nearly zero, whereas for iDPC imaging, it depends on the thickness[57]. The electron beam current utilized for HAADF-STEM and iDPC-STEM acquisitions was 30 pA. STEM-EELS experiments were conducted with a probe current of 70 pA, a collection angle of 100 mrad, a pixel time of 0.03 s, and a dispersion of 0.3 eV per channel. In the case of EDS measurements, the current was set at 300 pA for low magnification and 70 pA for high magnification. No apparent beam damage or atom displacement was observed during the HAADF and iDPC experiments, as confirmed by capturing multiple images at the same location. Furthermore, the sample was examined both before and after EDS and EELS measurements due to the substantial electron dose and total exposure time associated with these procedures. No significant changes in contrast were noted. To evaluate the effect of electron dose on sample damage, we examined the O $K$ edge and Ni $L$ edge, which are sensitive to irradiation damage in oxides, during STEM-EELS acquisition. We conducted EELS mappings over the same region eight times under identical experimental conditions, corresponding to an electron dose of approximately $4 \times 10^6$ e$^-$/Å$^2$ for each measurement. The averaged EELS signals are displayed in Supplementary Fig. S16. No significant modification of the O $K$ and Ni $L$ edges was observed. Therefore, we conclude that electron beam damage does not have a significant impact on the as-grown structures or introduce noticeable defects. 4D-STEM NBED data were acquired using a small convergence semi-angle 0.3 mrad, a camera length of 1.45 m, and an Electron Microscope Pixel Array Detector (EMPAD), the customized python scripts were utilized to select specific diffraction points for virtual imaging. TKD experiments were conducted using TEM samples within the FIB-SEM. The sample was positioned on a designated sample holder to ensure that the electron beam was penetrating perpendicularly to the sample surface. The acquisition conditions of the electron beam were 30 kV, 30 nA, and the step size was 5 nm.

### iDPC-STEM simulations

The iDPC-STEM simulations were performed using Dr. Probe software developed by Dr. Juri Barthel[58]. The imaging parameters are as follows: defocus value of 0 nm to −14 nm, accelerating voltage of 300 kV, convergence semi-angle of 25 mrad, and the collection angle ranging from 8 to 42 mrad.

## Data availability

The data that support the findings of this study are available from the corresponding author upon request.

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

## Acknowledgements

This work is financially supported by the National Key R&D Program of China (Grant No. 2022YFA1403200 (B.G. and H.W.) and No. 2022YFA1403603 (D.S and H.D.)), the National Natural Science Funds for Distinguished Young Scholar (52325105) (H.D.), the National Natural Science Foundation of China (No. 11874394 (B.G.), No. 52173215 (D.S.), No. 12061131001 (H.W.), No. A0402/11927809 (H.W.), No. 12204231 (Q.L.)), the National Natural Science Fund for Excellent Young Scientists Fund Program (Overseas) (D.S.), the Natural Science Foundation of Anhui Province for Excellent Young Scientist (2108085Y03) (D.S.), and the University Synergy Innovation Program of Anhui Province (No. GXXT-2020–003) (B.G.). The work made use of the resources of the Center for Electron Microscopy at Anhui University.

## Author contributions

D.S., B.G., and H.W. supervised the project. K.H. performed the experiment. Q.L. synthesized $Nd_{0.8}Sr_{0.2}NiO_2$. K.H. and Y.J. performed the statistical data. Z.L. completed the image simulation. K.H. and S.W. performed the TKD experiment. H.D. participated in guidance and discussions. K.H. and D.S. wrote the manuscript with input from all authors. All authors discussed the results and contributed to the manuscript.

## Competing interests

The authors declare no competing interests.
