## [peer review file · Nature Communications]

Atomic scale disorder and reconstruction in bulk infinite-layer nickelates lacking superconductivityREVIEWER COMMENTS

Reviewer #1 (Remarks to the Author):

In this submission, Kejun Hu et al. present a detailed investigation of the atomic structure of a topotactically-reduced bulk nickelate compound, using aberration-corrected scanning transmission electron microscopy (STEM). Infinite layer nickelates are currently a research subject of great interest, but also of ongoing debate. In particular, the exact conditions needed for infinite layer nickelates to be superconducting appears unsettled with, for instance, propositions of the importance of the film–substrate interface, strain control, H incorporation. Therefore, this study of bulk compounds is timely and, with revision, can in my evaluation be suitable for publication in Nature Communications.

From the analysis of various structural defects that deviate the lattice from being a “perfect” infinite layer structure, together with the detection of F impurity, the authors propose various possible reasons for the absence of superconductivity in the compound. In terms of experimental design, the manuscript focuses purely on structural characterization. While I find this to be of high quality – in particular the iDPC imaging – the scope excludes incorporation of numerical calculations or simulations to support the authors' arguments concerning the potential influence of various defects on the electronic behavior of the material under study. Instead, the authors effectively rely on inferring the electronic consequences from supporting references. This limited scope is not necessarily a problem. However, in the manuscript's current form I do find that the messaging outcome is rather “messy” and risks confusing the reader, because, out of the various possible reasons for absence of superconductivity, we have no idea of the magnitude of the effects. For instance, is a specific defect going to be a “deal breaker” by definitively eliminating the possibility of superconductivity, or will it instead have a more subtle effect? Equally, if superconductivity is to be achieved in a bulk compound, does that require eliminating all of the observed defects, or only some of them? After all, my understanding is that the first topotactically-reduced nickelate films to show superconductivity themselves contained numerous structural defects. If the authors can address these aspects, I believe it will make the manuscript more focused, easier to follow, and more relevant.

Having discussed results interpretation, I turn to the structural characterization. As remarked above, the authors present a beautiful quality of results, particularly with the iDPC measurements. However, I find that there is a major deficit that, I believe, the authors should address: characterization of the $\text{Ni}_{0.8}\text{Sr}_{0.2}\text{NiO}_3$ compound *before* the topotactic reduction is carried out. The reason for this is straightforward. Various structural defects are observed in the reduced compound. It is valuable to know if these existed in analogous forms before the reduction is carried out. (By analogous I mean, for instance Ruddlesden-Popper (RP) rock salt layers before as compared to fluorite structure layers after). Is the “block-like” structure already present? Does this derive from a 3D arrangement of RP defects? Having this information will enable the authors to give recommendations about which part of the synthesis needs to be improved in order to (potentially) obtain superconductivity in bulk nickelate compounds.

Having outlined two major aspects of the manuscript that I believe should be addressed for publication, I now detail more minor points that should be improved.

1. When read carefully, line-by-line, I find that the Introduction is hard to follow, particularly for a general reader who does not have a specialized background in infinite-layer compounds. Partly I think that this is because of some word choices and grammar, which I propose could be improved with the aid of a native English speaker or specialist. Examples are the “particularly” in line 49, the “could” in line 51, or lines 69-70 “the absence of superconductivity in bulk nickelates remains elusive”. (Do they mean “reasons remain elusive for the absence of superconductivity in bulk nickelates”?) However, it is also because of a lack of details and poor ordering of information. For instance, the meaning of T'-phase in line 56 needs to be clarified. From what I can tell, References 21-24 which are cited do not even mention the phrase “T'-phase”. Lines 62-63 circle back to this, and provides more clarification – however this ordering of information does not make sense. Further, no explanation or schematic illustration is given for what is meant by a “square-planar” structure, or “T'-phase”, nor do the authors specify to what unit cell type or symmetry group it

belongs. Moreover, by the end of the introduction, I do not understand if the infinite-layer structure or the "T'-phase" is preferred for superconductivity, or the exact relationship between the two with regards to superconductivity.

2. Fig. 1 shows that the polycrystalline sample contains many grain boundaries and voids. What role do these play in the electronic properties of the bulk sample? Do the electronic measurements control for their effects?

3. In the Results section, the authors refer to lattice directions [100], [010], [001]. They need to specify what setting or unit cell these refer to. Is it pseudo-cubic? Tetragonal? Ignoring defects, the starting nickelate compound is orthorhombic. How do the specified lattice directions relate to the initial orthorhombic cell?

4. Are the Ni-O planes formed by topotactic reduction uniformly aligned across the whole of one bulk grain (perhaps tied to one specific plane in the initial orthorhombic cell), or can they flip orientation within the grain?

5. Assuming that the initial compound, before topotactic reduction, contains RP defects, do these RP defects lie on a specific orthorhombic plane? If not, why are the T'-phase boundaries in the reduced compound all aligned with each other? This connects to comment #10 below, asking whether RP defects aligned on other planes lead to the formation of the "stripe" defects?

6. Line 121-122: why "pristine"? What is meant by "additional oxygen atoms" – additional to what?

7. Line 125: what is meant by "inserted" – was the oxygen layer inserted during reduction?

8. Figure 2(c) describes a mixture of T'-phases and infinite layer compound. Presumably, T'-phases are, like RP phases, a homologous compound series, each consisting of a certain number of infinite layers between two fluorite-structure layers. Therefore, in their labeling, what cut-off in numbers of layers do the authors use for deciding whether that part of the sample is T' or infinite layer?

9. Line 181: "imperfects" - > "imperfections"

10. Line 196: is the "stacking fault" only associated with a $\frac{1}{2}$ c shift of the lattice? Or is there also a lattice shift in the direction of the beam path? Parts (e) and (f) of Fig. 4 suggest that there is also a $\frac{1}{2}$ a shift of the lattice at these "stripe defects". These stripe defects appear to be deficient in Ni (different to an anti-phase boundary). Could their origin be RP defects in the initial compound that lie perpendicular to the eventual Ni-O planes of the reduced compound? This takes me back to the point, that it is vital to characterize the compound before the reduction is made. Finally, I am used to use of the phrase "stacking faults" with respect to compounds with close-packed structures and errors in their stacking sequences – is it the most appropriate term for these defects?

11. STEM methods. Were the iDPC and HAADF images recorded at the same or different defocus values? What filtering was applied to the iDPC image reconstruction from the four segment images? Which software was used for their acquisition and processing?

12. iDPC-STEM simulations: Juri Barthel's 2018 Ultramicroscopy paper should be appropriately cited. Line 329 refers to a defocus of -6 nm, but Fig. S4 presents a tableau going from a defocus of 0 μ m to -14 nm... The images in the tableau are too small to see details well.

Reviewer #2 (Remarks to the Author):

Hu, et al report atomic-scale characterization of defects in bulk RNiO₂ crystals and try to connect their observations with the lack of superconductivity (SC) observed in these compounds. The authors nicely catalogue and characterize stacking faults in the reduced phases, but I am not convinced that these secondary inclusions and stacking faults are the primary explanation for a

lack of SC in these samples. Many (nearly all) of the SC thin film samples also have a high (in fact, often higher) density of similar stacking faults, yet still exhibit clear signatures of SC. The presence of non-superconducting layers or inclusions also does not necessarily prevent superconductivity in this kind of layered sample. In principle, if there are regions which are sufficiently IL-like, then inclusions of secondary T' phases with the incorrect Ni filling could presumably just exist as passive non-superconducting layers, fundamentally no different from the inert substrate or capping layers in thin film samples. Similar effects are demonstrated in other layered oxide superconductors, eg. Sr_2RuO_4 (see APL Materials 10:041114, 2022). Rather, the segregation of Sr near fluorite layers may indicate that the effective doping within IL regions is too low for superconductivity. If the O-K and Ni-L edge EELS of the IL regions are consistent with other SC samples, then this does not yet explain the lack of observed SC.

Although their observation of chemical segregation and local distortions may be useful to other nickelate groups, their connection to superconductivity feels at this point a bit speculative. I would think that significantly more concrete insights could be gleaned at the mesoscale, for instance by considering the distribution of crystalline orientation at the tens or hundreds of nm length scales, e.g. across the fields of view shown in Figure 1 c-e. For instance, the authors point out “blurring” of the peaks in their SAED patterns, but these could also possibly be interpreted as bi-directional splitting of the peaks if the scattering area contained two domains with orthogonally oriented c-axes, the possibility of which is not discussed here. In fact, the peaks in figure 1e suggest either twin domains or a mostly unreduced phase: while strong anisotropy is seen in 1d between the 001 and 010 peaks, the pattern in 1e shows nearly isotropic lattice vectors in the 001 and 010 peaks. This distinction and a distribution of such twins could be clearly mapped out with scanning diffraction measurements and would be very useful to the field. Alternatively, atomic-scale characterization in different sub-regions of area 1e (i.e., across only horizontal or only vertical stripe regions (green boxes in attached) with comparisons of the local lattice orientation would also be useful.

In addition to the above suggestion to significantly improve the scope and impact of their work, I have some technical questions and general comments below:

Technical questions:

1. What are the characteristic domain sizes of the precursor as compared to the reduced crystals?
2. Lines 170-172: O-K edge EELS of the IL and T' phases: where precisely are the different spectra collected from? Do these regions include summing over the fluorite layers, or only the regions between them?
3. The pre-edge region in the O-K EEL spectrum indicates a poor background fit/subtraction. Can the authors improve this to obtain a flat, zero-values pre-edge?

General:

Overall, I found the results of the paper presented clearly, but the introduction is not very coherent and should be revised for clarity and accuracy. There are several instances of the authors trying to tie together unrelated conclusions and studies which is a bit confusing, for example:

4. Paragraph 1 seems mostly about the d electron filling, but also tries to bring in the dimensionality and cites two papers which are only secondarily concerned with the formal d filling but rather more with the parent Mott-Hubbard electronic landscape (Hepting) and its evolution with added holes (Goode). The trilayer sample (Li) doesn't fall within the superconducting dome. These are all important references certainly worth including, but the relation of each to its discussion in the paper could be improved.
5. Line 59 refers to “different ionic radii” as the reason for RP faults in thin films. Actually there are two distinct challenges being conflated here: the RP faults arise mostly because of the high tensile strain in precursor films grown on substrates which are optimized for the IL phase (doping has some subtle contribution here, but far more important is the primary R radius, e.g. La vs Nd (see APL Materials 8:4, 041107, 2020; Advanced Materials, 2104083, 2021; Nature Communications 14:1468, 2023). The extra difficulty for doped films relates also in large part to the high Ni valence which must be adopted in the doped precursor (see Sci. Adv. 9, eadh3327, 2023).
6. Discussion of various RP phases: the RP defects which form in precursor films under tensile strain are distinct from the layered RP compounds which are intentionally grown to stabilize intermediate nickel valences. The bilayer La compounds under pressure are so far understood to

be quite different from the reduced layered compounds.

Minor comments:

7. In the abstract line 35: I think the "even" should be removed: by definition, a fully reduced film will have no residual apical oxygens, so the comment refers to the (very common) issue of partially incomplete reduction.

8. In line 52: "confirmed" seems strong given H-free reduction of superconducting samples (see Phys. Rev. Materials 7, 013802, 2023) and the lack of confirmation by other groups of the H doping effect. I would recommend replacing with "suggested", "proposed", or "explored". In fact, relating to my previous comments, it could simply be removed as the importance or not of H does not seem related to the conclusion of quasi-2D-like structure (which is anyways implied by the crystal structure regardless of H impurities).

9. In line 122: "additional oxygen atoms" in reduced RP fluorite layers are not actually additional, these are the same O which were already present in the rock salt, but the configuration has changed. The compound has overall fewer O than in the precursor phase.

Reviewer #3 (Remarks to the Author):

The paper discusses how microstructure and local atomic arrangements in Nd_{0.8}Sr_{0.2}NiO₂ can lead to a non-superconducting state. These atomic scale observations range from point defects of F to remaining (non-reduced) oxygen sites. The authors do a good job relating these observations back to prior works, building upon a preexisting framework of knowledge and leave plenty for future researchers to follow up on. The techniques are adequate and appropriate for the purpose of the current paper. I quite enjoy the variety of techniques used, ranging from classic diffraction to newer imaging, to high resolution EELS. The figures are altogether attractive, and the subject of the manuscript is a "hot topic", so Nature Communication is an appropriate journal. There are a few points that should be addressed to strengthen the validity of certain claims. I have also provided a few suggestions for modified analysis that would provide more reliable metrics and statistics.

1. In Figure 1, the streaks mentioned in the diffraction pattern are not visible. Zoom in to show the first (an maybe second) order Bragg peaks to make things more clear. Add the full diffraction as supporting if you like.

2. In Figure 2, I assume that *eyy* is the [001] axis. This should be clarified. I also encourage the author to not use GPA. They have beautiful images from which the atom positions can be directly extracted. From the positions they can map average bond lengths in each row directly and provide error bars. GPA can be subject to artifacts. The same should be done for Fig 4.

GPA artifacts: <https://doi.org/10.1016/j.ultramic.2015.05.020>

iDPC/HR-STEM profiles with error bars:

<https://doi.org/10.1038/s41586-021-04238-z>, <https://doi.org/10.1002/adma.202207736>,

<https://doi.org/10.1016/J.ULTRAMIC.2017.06.002>

3. In Figure 3, are the EDS atomic resolution? It does not appear so. The data is good, but stating that these are high-resolution is an overreach.

4. Regarding, "The larger ratio of L3/L2 corresponds to the higher valence of Ni". There is a common misconception that the L3/L2 ratio results from valence state changes. The ratio is highly correlated with valence state. The real underlying reason for the ratio is crystal field splitting that or electron correlation. Crystal field splitting results from changes in coordination, which is in turn related to valence state and why L3/L2 trends so strongly. It would be best to state the correct physics regarding the coordination change then make it clear that a change in valence state is inferred.

5. "Remarkably, there is still residual apical oxygen in Nd/Sr plane after topotactic reduction, as indicated by the orange arrows in Fig. 5c - 5e with a random distribution." Do you mean "Remarkably, there is still a **random distribution of** residual apical oxygen in Nd/Sr plane after topotactic reduction, as indicated by the orange arrows in Fig. 5c - 5e."

6. You make a strong point that the materials are sensitive to defects. Most nickelates are prone to damage from sample preparation and the electron beam. How have you determined that the FIB

Ga-beam or the electron beam have not resulted in the observed structure and lack of atomic sites?

RE: Manuscript Number: NCOMMS-23-54994

“Atomic origin of absent superconductivity in bulk infinite-layer nickelate” by Kejun Hu et al.

Response to reviewers

We would like to express our gratitude to the reviewers for their diligent examination and insightful feedback on our work. The revised manuscript has been crafted, incorporating all the comments and suggestions provided by the reviewers. We look forward to continued support from the reviewers.

For your convenience, detailed point-to-point responses to all raised comments are presented below. All page numbers, references, and figures correspond to the revised version of the manuscript. Changes have been indicated in red throughout the revised manuscript.

Sincerely yours

Dongsheng Song on behalf of all of the authors

Institutes of Physical Science and Information Technology

Anhui University, Hefei 230601, China

E-mail: dsong@ahu.edu.cn

Responses to the comments raised by the referees

Reviewer #1:

In this submission, Kejun Hu et al. present a detailed investigation of the atomic structure of a topotactically-reduced bulk nickelate compound, using aberration-corrected scanning transmission electron microscopy (STEM). Infinite layer nickelates are currently a research subject of great interest, but also of ongoing debate. In particular, the exact conditions needed for infinite layer nickelates to be superconducting appears unsettled with, for instance, propositions of the importance of the film–substrate interface, strain control, H incorporation. Therefore, this study of bulk compounds is timely and, with revision, can in my evaluation be suitable for publication in Nature Communications.

Response:

We thank the referee for the precise judgment of the merit of our work and positive comments which allowed us to improve the paper. The revisions have been made point by point in the following to further improve the manuscript.

Major Comments:

Comment 1: *From the analysis of various structural defects that deviate the lattice from being a “perfect” infinite layer structure, together with the detection of F impurity, the authors propose various possible reasons for the absence of superconductivity in the compound. In terms of experimental design, the manuscript focuses purely on structural characterization. While I find this to be of high quality – in particular the iDPC imaging – the scope excludes incorporation of numerical calculations or simulations to support the authors' arguments concerning the potential influence of various defects on the electronic behavior of the material under study. Instead, the authors effectively rely on inferring the electronic consequences from supporting references. This limited scope is not necessarily a problem. However, in the manuscript's current form I do find that the messaging outcome is rather “messy” and risks confusing the reader, because, out of the various possible reasons for absence of superconductivity, we have no idea of the magnitude of the effects. For instance, is a specific defect going to be a “deal breaker” by definitively eliminating the possibility of superconductivity, or will it instead have a more subtle effect? Equally, if superconductivity is to be achieved in a bulk compound, does that require eliminating all of the observed defects, or only some of them? After all, my understanding is that the first topotactically-reduced nickelate films to show superconductivity themselves contained numerous structural defects. If the authors can address these aspects, I believe it will make the manuscript more focused, easier to follow, and more relevant.*

Response:

We appreciate the reviewer's constructive comments and suggestions. We agree with the reviewer that the various possible reasons for the absence of superconductivity should be clarified more clearly. Based on our findings, we think that the three following aspects may play the dominant roles in determining the superconductivity in bulk samples:

1. The primary condition for the manifestation of superconductivity in nickelates is their metallic character, which is not observed in bulk samples displaying insulating behavior. This observation

may be ascribed to two main factors. Firstly, numerous stripe phases, such as the T'-type phase ($R_{n+1}Ni_nO_{2n+2}$), are present within the grains exhibiting insulating behavior [*Nat. Mater.* 21, 160-164, 2022.]. These phases originate from the parent RP phase of the $Nd_{0.8}Sr_{0.2}NiO_3$ compound before undergoing topotactic reduction (as experimentally demonstrated in **Comment 2** below). Although the T'-type phase of $Nd_6Ni_5O_{12}$ compound with a layer number of $n = 5$ exhibits superconductivity through self-doping [*Nat. Mater.* 21, 160-164, 2022.], the layer number varies in the bulk case. Secondly, proper doping is essential for conferring metallic character and, consequently, inducing superconductivity. As discussed previously [*Phys. Rev. Lett.* 125, 027001, 2020., *Phys. Rev. Mater.* 4, 121801, 2020., *Phys. Rev. Lett.* 125, 147003, 2020., *Proc. Natl Acad. Sci. USA* 118, e2007683118, 2021.], the pure infinite-layer phase does not display superconductivity. Hence, despite the existence of Sr everywhere in the grains of the infinite-layer phase in the bulk sample, Sr doping level may be lower compared to that required for the superconducting phase. This discrepancy is attributed to the segregation of Sr across the interface of the T'-type phase, resulting in a possible low Sr content in the infinite layer, as shown in Fig. 3. The lower concentration of Sr doping is also revealed by the reduced intensity of pre-peak of the O K edge in Supplementary Fig. S7, consistent with previous results [*Proc. Natl Acad. Sci. USA* 118, e2007683118, 2021.]. Consequently, achieving superconductivity in bulk nickelates necessitates the preparation of a high-quality $Nd_{0.8}Sr_{0.2}NiO_3$ phase as the initial step. In cases where the RP phase cannot be completely eliminated, a higher concentration of Sr doping might be necessary to prevent underdoping caused by Sr segregation in the reduced RP phase (i.e., T'-type phase).

2. As noted by the reviewer, many superconducting thin films exhibit a high density of defects akin to those found in bulk materials. However, these defects, such as non-superconducting layers or inclusions, are typically confined to the upper regions of the thin films. In contrast, the superconducting layer with the infinite layer structure and proper hole doping near the substrate tend to be well-preserved throughout the sample or within a large area. This preservation ensures the presence of the superconductivity signature in electrical measurements. In the case of bulk materials, although infinite layer phases are indeed present, the continuity of the NiO_2 planes within the grains is disrupted by numerous randomly distributed stack faults and interleaved structures. Furthermore, these striped infinite-layer phases even undergo orientation flips within the grain, as illustrated in Supplementary Fig. S13. Establishing a superconducting channel becomes challenging under these conditions. Therefore, the elimination or reduction of these defects is imperative to achieve superconductivity in the bulk case, aligning with the requirements described in the previous paragraph.

3. In addition to the aforementioned issues, the incomplete removal of apical oxygen in $Nd_{0.8}Sr_{0.2}NiO_2$, as depicted in Fig. 5, introduces external structural distortion that impacts the flatness of the NiO_2 plane. This incomplete removal is considered a contributing factor to the absence of superconducting properties. The presence of residual oxygen atoms in the Nd/Sr plane disrupts the long-range order of the infinite-layer structure, leading to strong scattering and suppressing the superconductivity characterized by a low transition temperature, as discussed in a prior study on thin films [*Appl. Phys. Lett.* 123, 182601, 2023.]. Moreover, the insufficient topotactic reduction observed in the bulk sample is likely attributed to its large volume compared to thin films, which typically have a thickness of tens of nanometers. As a result, it is crucial to engineer the process of topotactic reduction to optimize the infinite layer and ultimately induce superconducting.

These discussions outlined above have been incorporated into the **Discussion** section of the revised manuscript. We hope that these additions will enhance the clarity of the manuscript, enabling both the reviewer and readers to follow the content more easily.

Revision:

The following statements have been involved in the Discussion part of the revised manuscript,

“The primary condition for the manifestation of superconductivity in nickelates is their metallic character, which is not observed in the bulk sample displaying insulating behavior. This may be induced by two main factors. Firstly, numerous stripe phases, such as the T'-type phase ($R_{n+1}Ni_nO_{2n+2}$), are present within the grains exhibiting insulating behavior³⁰. These phases originate from the parent RP phase of the $Nd_{0.8}Sr_{0.2}NiO_3$ compound before undergoing topotactic reduction, as shown in Supplementary Fig. S11 and S12. Then, these RP phases were transformed into the T'-type phase during the subsequent topotactic reduction process. Although the $Nd_6Ni_5O_{12}$ compound with a layer number of $n = 5$ exhibits superconductivity through self-doping³⁰, the layer number varies from place to place in the grain of bulk samples. Secondly, proper doping is essential for conferring metallic character and, consequently, inducing superconductivity. As discussed previously^{12-14,31}, the infinite-layer phase with a low hole doping level does not display superconductivity. Hence, despite the existence of a portion of the infinite-layer phase in the bulk sample with a certain amount of Sr, while the doping level may be lower compared to the threshold required for the superconducting phase. This can be corroborated by the observation of the segregation of Sr across the interface of the T'-type phase, resulting in a low Sr content in the infinite layer, as shown in Fig. 3. The lower concentration of Sr doping is also revealed by the reduced intensity of pre-peak of the O K edge in Supplementary Fig. S7, consistent with previous results³¹. Consequently, achieving superconductivity in bulk nickelates necessitates the preparation of a high-quality $Nd_{0.8}Sr_{0.2}NiO_3$ phase as the initial step. In cases where the RP phase cannot be completely eliminated, a higher concentration of Sr doping might be necessary to prevent underdoping caused by Sr segregation in the reduced RP phase (i.e., T'-type phase).

Also, it's important to note that many thin films, i.e. nickelates^{17, 20} and Sr_2RuO_4 ⁵⁵, exhibit superconductivity even in the presence of a high density of defects similar to those found in bulk materials. However, these defects, such as non-superconducting layers or inclusions, are usually confined to the upper regions of the thin films²⁰. In contrast, the superconducting layer with the infinite layer structure and proper hole doping near the substrate tend to be well-preserved throughout the sample or within a large area. Consequently, the signature of superconductivity is evident in electrical measurements. In the case of bulk materials, the infinite layer phases are indeed present, but the continuity of the NiO_2 planes within the grains is disrupted by numerous and randomly distributed stack faults and interleaved structures. Furthermore, these striped infinite-layer phases even undergo orientation flips within the grain, as illustrated in Supplementary Fig. S13. Establishing a superconducting channel becomes challenging under these conditions. Therefore, eliminating or reducing these defects is necessary for achieving superconductivity in the bulk case.

At last, the incomplete removal of apical oxygen in $Nd_{0.8}Sr_{0.2}NiO_2$, as depicted in Fig. 5, introduces external structural distortion that impacts the flatness of the NiO_2 plane. This makes the buckling of the NiO_2 planes, leading to enhanced scattering of electrons and inhibiting the pairing. Thus, this incomplete removal is considered to be the alternative possibility for the absence of

superconductivity in the present bulk samples. The presence of residual oxygen atoms in the Nd/Sr plane disrupts the long-range order of the infinite-layer structure, leading to suppressed superconductivity characterized by a low transition temperature, as discussed in a prior study on thin films⁵⁶. Moreover, the insufficient topotactic reduction observed in the bulk sample is likely attributed to its large volume compared to thin films, which typically have a thickness of tens of nanometers. As a result, it is crucial to engineer the process of topotactic reduction to optimize the infinite layer and ultimately induce superconducting.”

Comment 2: *Having discussed results interpretation, I turn to the structural characterization. As remarked above, the authors present a beautiful quality of results, particularly with the iDPC measurements. However, I find that there is a major deficit that, I believe, the authors should address: characterization of the $\text{Ni}_{0.8}\text{Sr}_{0.2}\text{NiO}_3$ compound before the topotactic reduction is carried out. The reason for this is straightforward. Various structural defects are observed in the reduced compound. It is valuable to know if these existed in analogous forms before the reduction is carried out. (By analogous I mean, for instance Ruddlesden-Popper (RP) rock salt layers before as compared to fluorite structure layers after). Is the “block-like” structure already present? Does this derive from a 3D arrangement of RP defects? Having this information will enable the authors to give recommendations about which part of the synthesis needs to be improved in order to (potentially) obtain superconductivity in bulk nickelate compounds.*

Response:

Thanks for the reviewer’s constructive comments. We strongly agree that characterizing the $\text{Nd}_{0.8}\text{Sr}_{0.2}\text{NiO}_3$ compound before topotactic reduction is essential to substantiate the origin of structural defects. Hence, we conducted a structural characterization of the $\text{Nd}_{0.8}\text{Sr}_{0.2}\text{NiO}_3$ compound before topotactic reduction, as illustrated in Supplementary Fig. S11 and Fig. S12 below. The nickelate parent phase inherently harbors numerous defective structures, including the RP phase. This cross-arrangement of defective structures constitutes a significant factor contributing to the formation of three-dimensional block-like structures. These results and discussions have been addressed in the Supplementary Information.

Revision:

The following discussions have been added in the Supplementary Information,

“In STEM characterization of bulk $\text{Nd}_{0.8}\text{Sr}_{0.2}\text{NiO}_3$ before topotactic reduction, inherent defect structures similar to the RP phase were observed in the grains of its parent phase (Supplementary Fig. S11). The characteristic domain size of the precursor $\text{Nd}_{0.8}\text{Sr}_{0.2}\text{NiO}_3$ ranges from tens to hundreds of nanometers in length and several to tens of nanometers in width. These defects likely contribute to the formation of a 3D block-like structure within grains after reduction. The results of EDS mapping showed that the distribution of elements was uniform as that of the reduced samples, as shown in Supplementary Fig. S11a. This size of grains is comparable to that observed in the reduced samples. The magnifying bright-field (BF) image in Fig. S11b and Fig. S11d reveals the presence of numerous stripes within the $\text{Nd}_{0.8}\text{Sr}_{0.2}\text{NiO}_3$ grains, similar to the block-like structure in the reduced samples. The block-like configuration within the parent phase consists of alternating RP phases, each with a distinct *c*-axis orientation, as illustrated in Supplementary Fig. S12. Consequently, the defect structures present in these parent phases serve as the origin of the block-

like structure observed in the $\text{Nd}_{0.8}\text{Sr}_{0.2}\text{NiO}_2$ grains after following topotactic reduction.”

Supplementary Fig. S11 | Structure characterization of bulk $\text{Nd}_{0.8}\text{Sr}_{0.2}\text{NiO}_3$ before topotactic reduction. **a** EDS mapping of $\text{Nd}_{0.8}\text{Sr}_{0.2}\text{NiO}_3$ polycrystalline at low magnification. **b** and **d** High-resolution BF-STEM image of $\text{Nd}_{0.8}\text{Sr}_{0.2}\text{NiO}_3$. **c** and **e** The magnified HAADF-STEM images of marked regions in **b** and **d**.

Supplementary Fig. S12 | Atomic structure in bulk $\text{Nd}_{0.8}\text{Sr}_{0.2}\text{NiO}_3$ before topotactic reduction. **a** Atom model of RP phase ($n = 4$). **b** Atomic-resolved HAADF-STEM image of **Fig. S11c**. **c** and **d** The ϵ_{xx} and ϵ_{yy} strain maps are calculated by GPA for **b**. **e** Atom model of RP phase with c -axis along electron beam direction (the oxygen atoms are not shown here). **f** Atomic-resolved HAADF-STEM image of **Fig. S11e**. **g** and **h** The ϵ_{xx} and ϵ_{yy} strain maps are calculated by GPA for **f**.

Minor Comments: Having outlined two major aspects of the manuscript that I believe should be addressed for publication, I now detail more minor points that should be improved.

Question 1. *When read carefully, line-by-line, I find that the Introduction is hard to follow, particularly for a general reader who does not have a specialized background in infinite-layer compounds. Partly I think that this is because of some word choices and grammar, which I propose could be improved with the aid of a native English speaker or specialist. Examples are the “particularly” in line 49, the “could” in line 51, or lines 69-70 “the absence of superconductivity in bulk nickelates remains elusive”. (Do they mean “reasons remain elusive for the absence of superconductivity in bulk nickelates”?) However, it is also because of a lack of details and poor ordering of information. For instance, the meaning of T'-phase in line 56 needs to be clarified. From what I can tell, References 21-24 which are cited do not even mention the phrase “T'-phase”. Lines 62-63 circle back to this, and provides more clarification – however this ordering of information does not make sense. Further, no explanation or schematic illustration is given for what is meant by a “square-planar” structure, or “T'-phase”, nor do the authors specify to what unit cell type or symmetry group it belongs. Moreover, by the end of the introduction, I do not understand if the infinite-layer structure or the “T'-phase” is preferred for superconductivity, or the exact relationship between the two with regards to superconductivity.*

Response 1:

Thanks for the reviewer’s comments. The **Introduction** section has been extensively rewritten (see the revision below). The issues you highlighted above have been addressed as follows,

“particularly” in line 49, the “could” in line 51, or lines 69-70 “the absence of superconductivity in bulk nickelates remains elusive”. (Do they mean “reasons remain elusive for the absence of superconductivity in bulk nickelates”?)

We have deleted “particularly” in line 48 and “could” in line 50 to ensure a smoother, more logical text.

The sentence “the absence of superconductivity in bulk nickelates remains elusive” has been revised as “the reasons for the absent superconductivity in bulk nickelates remain elusive”.

However, it is also because of a lack of details and poor ordering of information. For instance, the meaning of T'-phase in line 56 needs to be clarified. From what I can tell, References 21-24 which are cited do not even mention the phrase “T'-phase”. Lines 62-63 circle back to this, and provides more clarification – however this ordering of information does not make sense.

In order to describe the T'-type phase more clearly, we have added more descriptions in the **Introduction** section (see the revision below) and added appropriate references in lines 64-69.

Further, no explanation or schematic illustration is given for what is meant by a “square-planar” structure, or “T'-phase”, nor do the authors specify to what unit cell type or symmetry group it belongs.

The term “square-planar” means “quasi-two-dimensional NiO₂ planes in a square-planar coordination [Nat. Mater. 21, 160-164, 2022.]” in line 68, and a description of the T'-type phase as belonging to the tetragonal crystal system has been added in line 65.

Moreover, by the end of the introduction, I do not understand if the infinite-layer structure or the “T'-phase” is preferred for superconductivity, or the exact relationship between the two with regards to superconductivity.

Currently, superconductivity has been observed among thin films featuring infinite-layer structured $R_{1-x}A_x\text{NiO}_2$ (where $R = \text{La, Pr, Nd}$, and $A = \text{Sr, Ca, Eu}$). However, thin films of the T'-type phases do not typically exhibit superconductivity, except for $\text{Nd}_6\text{Ni}_5\text{O}_{12}$ with $n = 5$, which meets the criterion where the outer electrons $d^{8.8}$ fall within the superconducting range [*Nat. Mater.* 21, 160-164, 2022.].

Revision:

We have rewritten the Introduction in the revised manuscript,

“The emergence of superconductivity in nickelates represents a pivotal breakthrough in the exploration of superconducting materials and mechanisms¹, which is underscored by the notable similarity in the atomic configuration and electronic characteristics between the infinite NiO_2 layers in nickelates and the CuO_2 planes in cuprates²⁻⁶. The initial discovery occurred in $\text{Nd}_{0.8}\text{Sr}_{0.2}\text{NiO}_2$, achieved through the topotactic chemical reduction of the perovskite precursor $\text{Nd}_{0.8}\text{Sr}_{0.2}\text{NiO}_3$ using CaH_2 ⁷⁻¹¹. Subsequently, other nickelates thin films with diverse compositions, such as $R_{1-x}A_x\text{NiO}_2$ ($R = \text{La, Pr, Nd}$, and $A = \text{Sr, Ca, Eu}$), have demonstrated superconductivity under appropriate doping concentrations¹²⁻¹⁹.

Two crucial factors contributing to the observed superconductivity are the infinite layer structure and proper hole doping achieved through the alkaline earth substitution²⁰. The infinite-layer structure is typically derived from the precursor perovskite phase. However, synthesizing a pure perovskite phase, i.e., $\text{Nd}_{0.8}\text{Sr}_{0.2}\text{NiO}_3$, remains challenging due to structural instability²⁰. This instability is primarily attributed to the high tensile strain in precursor films grown on substrates¹⁹⁻²¹ and the instability of the high-valence state $\text{Ni}^{18,22}$. Consequently, the Ruddlesden-Popper phase (RP, $R_{n+1}\text{Ni}_n\text{O}_{3n+1}$, R is rare-earth) is often observed in nickelates thin films, hindering the manifestation of superconductivity^{23,24}. Recent developments even show that intentionally grown layered RP phase $\text{La}_3\text{Ni}_2\text{O}_7$ can exhibit superconductivity with a T_c of 80 K under pressure²⁵. However, this superconducting mechanism differs from that of infinite-layer nickelates by stabilizing intermediate nickel valences^{25,26}. These orthorhombic RP phases ($R_{n+1}\text{Ni}_n\text{O}_{3n+1}$) typically transform into the tetragonal T'-type phase ($R_{n+1}\text{Ni}_n\text{O}_{2n+2}$) through oxygen deintercalation and structural rearrangement after topotactic reduction^{10,27}. The T'-type phase possesses quasi-two-dimensional NiO_2 planes in square-planar coordination with rare-earth fluorite blocking slabs interleaved in every n nickel layer^{7,28-30}. Yet, only a specific case, $\text{Nd}_6\text{Ni}_5\text{O}_{12}$ with $n = 5$, has been found to exhibit superconductivity with outer electrons $d^{8.8}$ ³⁰. These intricate phenomena manifest the strong correlation between the local atomic structure and superconductivity in nickelates. For the hole doping mechanism, the electronic structure measurements have provided insights within the Mott-Hubbard regime in the parent phase NdNiO_2 ⁶. The doping evolution has been convincingly demonstrated through locally resolved electron energy-loss spectroscopy (EELS)³¹. The interfacial reconstruction has been revealed to stabilize the external crystal structure and alleviate polar discontinuity in $\text{Nd}_{0.8}\text{Sr}_{0.2}\text{NiO}_2$ thin films³²⁻³⁵. Despite significant strides in addressing these key issues and advancing our understanding of superconductivity in nickelates thin films, the reasons

for the absent superconductivity in bulk nickelates remain elusive³⁶⁻³⁸.”

Question 2. *Fig. 1 shows that the polycrystalline sample contains many grain boundaries and voids. What role do these play in the electronic properties of the bulk sample? Do the electronic measurements control for their effects?*

Response 2:

Thanks for the reviewer's comments. The presence of grain boundaries and voids can indeed elevate resistance and deteriorate the insulating properties of the material. Typically, applying high pressure can mitigate the impact of grain boundaries on the transport properties of the material, resulting in increased density and reduced voids, as discussed in Li et al.'s study [*Commun. Mater.* 1, 16, 2020., the co-authors of the present manuscript]. However, despite the improvement in insulating properties through high-pressure application, superconducting properties remained elusive. This suggests that the presence of grain boundaries and voids does not have a detrimental effect on superconductivity. These discussions have been incorporated into the revised manuscript.

Revision 2:

**The following discussions have been added to the revised manuscript,
Page 6, Line 109-116,**

“The presence of grain boundaries and voids in bulk nickelate, as shown in Fig. 1b, can indeed elevate resistance and deteriorate the insulating properties of the material. As discussed in a previous study³⁶, applying high pressure can mitigate the impact of grain boundaries on the transport properties through increased density and reduced voids. However, despite the improvement in insulating properties under high-pressure conditions, superconducting properties remained elusive. This suggests that the presence of grain boundaries and voids does not have a detrimental effect on superconductivity.”

Question 3. *In the Results section, the authors refer to lattice directions [100], [010], [001]. They need to specify what setting or unit cell these refer to. Is it pseudo-cubic? Tetragonal? Ignoring defects, the starting nickelate compound is orthorhombic. How do the specified lattice directions relate to the initial orthorhombic cell?*

Response 3:

Thanks for the reviewer's comments. The parent phase $\text{Nd}_{0.8}\text{Sr}_{0.2}\text{NiO}_3$ has an orthorhombic structure with lattice parameters of $a_o = 5.39 \text{ \AA}$, $b_o = 5.38 \text{ \AA}$, and $c_o = 7.61 \text{ \AA}$. In the manuscript, we use the pseudocubic unit cell with lattice parameters of $a_{pc} = 3.808 \text{ \AA}$, $c = 3.805 \text{ \AA}$ to define the orientation. The orientation between the pseudocubic indices and the orthorhombic indices of the $\text{Nd}_{0.8}\text{Sr}_{0.2}\text{NiO}_3$ are $(010)_{pc} // (110)_O$, $(100)_{pc} // (110)_O$, $(001)_{pc} // (001)_O$.

The infinite layer $\text{Nd}_{0.8}\text{Sr}_{0.2}\text{NiO}_2$ has a tetragonal structure with $a_t = b_t = 3.92 \text{ \AA}$, $c_t = 3.34 \text{ \AA}$. In order to elucidate the relationship between the two lattice orientations before and after the topotactic transformation, the atomic structure model with direction vectors is illustrated in Figure 1a in the revised manuscript.

The information has been incorporated into the revised manuscript.

Revision 3:

We have added the following statements in the revised manuscript,

Page 5, Line 100-106,

“The parent phase $\text{Nd}_{0.8}\text{Sr}_{0.2}\text{NiO}_3$ has an orthorhombic structure with lattice parameters of $a_o = 5.39$ Å, $b_o = 5.38$ Å, and $c_o = 7.61$ Å, the pseudocubic unit cell with lattice parameters of $a_{pc} = 3.808$ Å, $c = 3.805$ Å to define the orientation. The orientation between the pseudocubic indices and the orthorhombic indices of the $\text{Nd}_{0.8}\text{Sr}_{0.2}\text{NiO}_3$ are $(010)_{pc} // (110)_O$, $(100)_{pc} // (110)_O$, $(001)_{pc} // (001)_O$. The infinite layer $\text{Nd}_{0.8}\text{Sr}_{0.2}\text{NiO}_2$ has a tetragonal structure with $a_t = b_t = 3.92$ Å, $c_t = 3.34$ Å. The atomic models before and after the topotactic transformation are illustrated in Fig. 1a.”

Figure 1 has been updated in the revised manuscript as shown below.

Fig. 1 | Polycrystalline $\text{Nd}_{0.8}\text{Sr}_{0.2}\text{NiO}_2$ imaged by HAADF-STEM. **a** Atomic structure model with direction vectors of perovskite RNiO_3 and infinite layer RNiO_2 (R : Nd/Sr). **b** EDS mappings of $\text{Nd}_{0.8}\text{Sr}_{0.2}\text{NiO}_2$ at low magnification. **c** Temperature dependence of resistivity for bulk $\text{Nd}_{0.8}\text{Sr}_{0.2}\text{NiO}_2$ in the range of 2-300 K. **d** High-resolution HAADF-STEM image of $\text{Nd}_{0.8}\text{Sr}_{0.2}\text{NiO}_2$. **e** and **f** The magnified images of marked regions in **d**, with the corresponding SAED pattern in the insets.

Question 4. *Are the Ni–O planes formed by topotactic reduction uniformly aligned across the whole of one bulk grain (perhaps tied to one specific plane in the initial orthorhombic cell), or can they flip orientation within the grain?*

Response 4:

Thanks for the reviewer’s comment. The NiO_2 planes after topotactic reduction are not uniformly aligned across the whole grains. As indicated by the yellow and red boxes in Supplementary Fig. S13, the NiO_2 planes are observed with the orientation in two perpendicular directions, disrupting the continuity of the NiO_2 planes. This information has been incorporated into the **Discussion** section in the revised manuscript.

Supplementary Fig. S13 | Discontinuity of NiO₂ planes within the grains. **a** and **b** The red and yellow dotted boxes correspond to domains with the orientations of [100] and [001], respectively.

Revision 4:

We have added the following statements in the revised manuscript,

Page 19, Line 375-377,

“Furthermore, these striped infinite-layer phases even undergo orientation flips within the grain, as illustrated in Supplementary Fig. S13.”

Question 5. *Assuming that the initial compound, before topotactic reduction, contains RP defects, do these RP defects lie on a specific orthorhombic plane? If not, why are the T'-phase boundaries in the reduced compound all aligned with each other? This connects to comment #10 below, asking whether RP defects aligned on other planes lead to the formation of the “stripe” defects?*

Response 5:

Thanks for the reviewer’s comment. After checking the compound before topotactic reduction, the RP defects are found to lie along the *c*-axis by examining many regions. As a result, the T'-type phase boundaries in the reduced compound align with each other along the *c*-axis. We did not observe the formation of RP defects on other planes in our experiments.

However, the *c*-axis of the RP phase in the parent phase is randomly distributed and forms a 3D bock-like structure as shown in Supplementary Fig. S11 for the sample before topotactic reduction. Consequently, they could be perpendicular to each other, giving rise to two perpendicular T'-type phases after reduction, as illustrated in Fig. 1f. Alternatively, they could align in the same direction, yet with domains on the left and right sides shifting both in the [100] and [001] directions. This could lead to the creation of a stacking-fault-like plane after reduction, as depicted in Fig. 4b at **#Comment 10**.

Question 6. *Line 121-122: why “pristine”? What is meant by “additional oxygen atoms” – additional to what?*

Response 6:

Thanks for the reviewer's comment. We apologize for the confusion caused by the wording in the manuscript. "pristine" refers to the RP phases that are not yet reduced by CaH_2 .

"additional oxygen atoms" refers to the oxygen atoms between two R layers as highlighted in red font in Fig. 2a.

We have removed these misleading terms from the revised manuscript.

Revision 6:

We have changed the following statements in the revised manuscript,

Page 8, Line 160,

The sentence "Moreover, the pristine rock salt layers are changed into the fluorite structure after topotactic reduction, with additional oxygen atoms inserted between two R layers as indicated by the red spheres in Fig. 2a." has been revised as "Furthermore, the rock salt layers undergo a transformation into the fluorite structure after topotactic reduction, involving the rearrangement of oxygen atoms between two R layers, as illustrated by the red spheres in Fig. 2a."

Question 7. *Line 125: what is meant by "inserted" – was the oxygen layer inserted during reduction?*

Response 7:

Thanks for the reviewer's comment. We apologize for the confusion caused by the wording in the manuscript. The oxygen layer is formed through the rearrangement of oxygen atoms during the reduction process rather than being inserted. To prevent any potential misinterpretation, the sentence "The inserted oxygen layer is indicated by orange arrow." has been revised as "The rearranged oxygen layer is indicated by orange arrow."

Question 8. *Figure 2(c) describes a mixture of T' -phases and infinite layer compound. Presumably, T' -phases are, like RP phases, a homologous compound series, each consisting of a certain number of infinite layers between two fluorite-structure layers. Therefore, in their labeling, what cut-off in numbers of layers do the authors use for deciding whether that part of the sample is T' or infinite layer?*

Response 8:

Thanks for the reviewer's comment. As the reviewer said, the T' -type phase is composed of a certain number of infinite layers between two fluorite-structure layers. Hence, it is not definitive to specify a cut-off value. To distinguish between the T' -type phase and infinite layers in Fig. 2c, we simply use $n > 10$ as a rough criterion here. This point has been addressed in the revised manuscript.

Revision 8:

We have added the following statements in Fig. 2c of the revised manuscript,

Page 9, Line 183-184,

"To differentiate between T' -type phase and infinite layer structures, the latter are simply defined here as $n > 10$ layers."

Question 9. Line 181: “imperfects” - > “imperfections”

Response 9:

Thanks for the reviewer’s comment. “imperfects” has been corrected to “imperfections”.

Question 10. Line 196: *is the “stacking fault” only associated with a $\frac{1}{2}c$ shift of the lattice? Or is there also a lattice shift in the direction of the beam path? Parts (e) and (f) of Fig. 4 suggest that there is also a $\frac{1}{2}a$ shift of the lattice at these “stripe defects”. These stripe defects appear to be deficient in Ni (different to an anti-phase boundary). Could their origin be RP defects in the initial compound that lie perpendicular to the eventual Ni–O planes of the reduced compound? This takes me back to the point, that it is vital to characterize the compound before the reduction is made. Finally, I am used to use of the phrase “stacking faults” with respect to compounds with close-packed structures and errors in their stacking sequences – is it the most appropriate term for these defects?*

Response 10:

Thanks for the reviewer’s comments. We have revised the above issues you mentioned point by point to further improve the manuscript.

is the “stacking fault” only associated with a $\frac{1}{2}c$ shift of the lattice? Or is there also a lattice shift in the direction of the beam path? Parts (e) and (f) of Fig. 4 suggest that there is also a $\frac{1}{2}a$ shift of the lattice at these “stripe defects”.

We apologize for the inaccurate statement. The “stacking fault” shifts along both $\frac{1}{2}c$ and $\frac{1}{2}a$ (electron beam direction) directions. Consequently, the displacement of the left and right parts amounts to $\frac{1}{2}[101]$. This “stacking fault” exhibits a structure akin to the rock salt layer in the RP phase, characterized by a shear fault along $\frac{1}{2}[101]$ of the R-O planes.

To avoid misleading, the sentence “In contrast to the T'-type phases with additional oxygen/fluorine layers, the vertical stripes are formed just by a $\frac{1}{2}c$ shift of the infinite-layer phase (left) with respect to the other one (right) along the [001] axis, similar to the stacking fault structure.”

has been revised as “In contrast to the T'-type phases with oxygen/fluorine layers, the vertical stripes are formed by a $\frac{1}{2}[101]$ shift of the infinite-layer phase (left) with respect to the other one (right), similar to the rock salt layer in the RP phase.” in line 249 of the revised manuscript.

These stripe defects appear to be deficient in Ni (different to an anti-phase boundary). Could their origin be RP defects in the initial compound that lie perpendicular to the eventual Ni–O planes of the reduced compound? This takes me back to the point, that it is vital to characterize the compound before the reduction is made.

As noted by the reviewer, the rock salt layer in the RP phase is deficient in Ni. The parent phase RNiO_3 comprises numerous cross-arranged RP phase structures, suggesting that its intricate structure gives rise to the defect layers. As shown in Supplementary Fig. S12b above (**Major comment #2**), the topotactic reduction process triggers an oxygen removal reaction along the RP

phase in the horizontal direction, causing the vertically oriented RP phase to retain the Ni-deficient rock salt layer. This results in a rock salt layer perpendicular to the NiO₂ plane forms, as depicted in Fig. 4a. The new experiments characterizing the compound before reduction have been incorporated into the revised manuscript (**Major comment #2**).

Finally, I am used to use of the phrase “stacking faults” with respect to compounds with close-packed structures and errors in their stacking sequences – is it the most appropriate term for these defects?

Thank you for the reviewer's suggestion. In some cases, the rock salt layer can be considered as a type of stacking fault. To distinguish their differences, we have revised the term “stacking fault” to “stacking-fault-like” and updated the description of Fig. 4b accordingly.

Figure 4 has been updated in the revised manuscript as shown below.

Fig. 4 | 3D block-like configurations in Nd_{0.8}Sr_{0.2}NiO₂ grains. a Atomic resolution HAADF-STEM images of the parallel and vertical stripes along the [100] zone axis. The orange and green arrows point to the interface of parallel stripes and vertical stripes, respectively. **b** Atom model of the vertical stripe structure. The enlarged iDPC-STEM image of the vertical stripe is displayed below. **c** and **d** The ϵ_{xx} and ϵ_{yy} strain maps are calculated by GPA for **a**. **e** HAADF-STEM image of the regions containing several kinds of stripe structures along [001] zone axis, which is indicated by the green dotted line. In addition, the slight change of atomic column contrast in the yellow

dotted box may be caused by the inclined wedge interface. **f** Schematic diagram of the 3D atomic model of the central joint area in **e**, in which only 1 cell is drawn in the direction [010] in Area 3 due to space. The black dashed line represents the stacking-fault-like plane, and the red dashed line represents the fluorite layer of T'-type phases. **g** Schematic diagram of a 3D block-like model composed of randomly distributed stripe domains in the block.

Question 11. *STEM methods. Were the iDPC and HAADF images recorded at the same or different defocus values? What filtering was applied to the iDPC image reconstruction from the four segment images? Which software was used for their acquisition and processing?*

Response 11:

Thanks for the reviewer's comment. The iDPC and HAADF images were recorded at different defocus values. Typically, the optimal defocus value for HAADF imaging is nearly zero, whereas for iDPC imaging, it depends on the thickness, as detailed in our prior publication [*Ultramicroscopy* 246, 113686, 2023.]. The iDPC images were reconstructed from the four segment images, employing a high-pass filter to diminish low frequency information. Both data acquisition and processing were carried out using the commercial Velox software. These details have been involved in the Methods section of the manuscript.

Revision 11:

We have added the following statements in the revised manuscript,

Page 22, Line 439-443,

“The iDPC images were reconstructed from the four segment images, employing a high-pass filter to diminish low frequency information. Both data acquisition and processing were carried out using the commercial Velox software. The optimal defocus value for HAADF imaging is nearly zero, whereas for iDPC imaging, it depends on the thickness⁵⁷.”

Question 12. *iDPC-STEM simulations: Juri Barthel's 2018 Ultramicroscopy paper should be appropriately cited. Line 329 refers to a defocus of -6 nm, but Fig. S4 presents a tableau going from a defocus of 0 um to -14 nm... The images in the tableau are too small to see details well.*

Response 12:

Thanks for the reviewer's comment. Juri Barthel's paper has been appropriately cited for the iDPC simulations. Additionally, the range of defocus values has been corrected to “0 nm to -14 nm.”

To enhance the presentation of simulated iDPC images, specific plots with distinct thicknesses and defocus values have been enlarged for better visibility in Supplementary Fig. S10.

Supplementary Fig. S10 | iDPC images simulations of the infinite layer structure. a Simulated iDPC images with different thicknesses and defocus values based on the atomic model along [100] orientation. The red box shows the conditions with better imaging contrast. **b** To better show the details, some of the images are zoomed in. The atom model of the infinite layer structure is displayed on the right-hand side for comparison.

Reviewer #2:

Comment 1: *Hu, et al report atomic-scale characterization of defects in bulk RNiO₂ crystals and try to connect their observations with the lack of superconductivity (SC) observed in these compounds. The authors nicely catalogue and characterize stacking faults in the reduced phases, but I am not convinced that these secondary inclusions and stacking faults are the primary explanation for a lack of SC in these samples. Many (nearly all) of the SC thin film samples also have a high (in fact, often higher) density of similar stacking faults, yet still exhibit clear signatures of SC. The presence of non-superconducting layers or inclusions also does not necessarily prevent superconductivity in this kind of layered sample. In principle, if there are regions which are sufficiently IL-like, then inclusions of secondary T' phases with the incorrect Ni filling could presumably just exist as passive non-superconducting layers, fundamentally no different from the inert substrate or capping layers in thin film samples. Similar effects are demonstrated in other layered oxide superconductors, e.g. Sr₂RuO₄ (see APL Materials 10:041114, 2022). Rather, the segregation of Sr near fluorite layers may indicate that the effective doping within IL regions is too low for superconductivity. If the O-K and Ni-L edge EELS of the IL regions are consistent with other SC samples, then this does not yet explain the lack of observed SC.*

Response 1:

Thanks for the reviewer's positive comment. Concerning the absence of superconductivity in present bulk samples, we completely agree with the referee that it is not solitarily induced by the secondary inclusions (such as defects) and stacking faults. Instead, we outlined several possibilities which are responsible for the absence of superconductivity. This concern was also mentioned by the first referee. These possibilities are as follows:

1. The primary condition for the manifestation of superconductivity in nickelates is their metallic character, which is not observed in bulk samples displaying insulating behavior. This observation may be ascribed to two main factors. Firstly, numerous stripe phases, such as the T'-type phase ($R_{n+1}Ni_nO_{2n+2}$), are present within the grains exhibiting insulating behavior [*Nat. Mater.* 21, 160-164, 2022.]. These phases originate from the parent RP phase of the Nd_{0.8}Sr_{0.2}NiO₃ compound before undergoing topotactic reduction. Although the T'-type phase of Nd₆Ni₅O₁₂ compound with a layer number of $n = 5$ exhibits superconductivity through self-doping [*Nat. Mater.* 21, 160-164, 2022.], the layer number varies in the bulk case. Secondly, proper doping is essential for conferring metallic character and, consequently, inducing superconductivity. As discussed previously [*Phys. Rev. Lett.* 125, 027001, 2020., *Phys. Rev. Mater.* 4, 121801, 2020., *Phys. Rev. Lett.* 125, 147003, 2020., *Proc. Natl Acad. Sci. USA* 118, e2007683118, 2021.], the pure infinite-layer phase does not display superconductivity. Hence, despite the existence of Sr everywhere in the grains of the infinite-layer phase in the bulk sample, Sr doping level may be lower compared to that required for the superconducting phase. This discrepancy is attributed to the segregation of Sr across the interface of the T'-type phase, resulting in a possible low Sr content in the infinite layer, as shown in Fig. 3. The lower concentration of Sr doping is also revealed by the reduced intensity of pre-peak of the O K edge in Supplementary Fig. S7, consistent with previous results [*Proc. Natl Acad. Sci. USA* 118, e2007683118, 2021.]. Consequently, achieving superconductivity in bulk nickelates necessitates the preparation of a high-quality Nd_{0.8}Sr_{0.2}NiO₃ phase as the initial step. In cases where the RP phase cannot be completely eliminated, a higher concentration of Sr doping might be

necessary to prevent underdoping caused by Sr segregation in the reduced RP phase (i.e., T'-type phase).

2. As noted by the reviewer, many superconducting thin films exhibit a high density of defects akin to those found in bulk materials. However, these defects, such as non-superconducting layers or inclusions, are typically confined to the upper regions of the thin films. In contrast, the superconducting layer with the infinite layer structure and proper hole doping near the substrate tend to be well-preserved throughout the sample or within a large area. This preservation ensures the presence of the superconductivity signature in electrical measurements. In the case of bulk materials, although infinite layer phases are indeed present, the continuity of the NiO₂ planes within the grains is disrupted by numerous randomly distributed stack faults and interleaved structures. Furthermore, these striped infinite-layer phases even undergo orientation flips within the grain, as illustrated in Supplementary Fig. S13. Establishing a superconducting channel becomes challenging under these conditions. Therefore, the elimination or reduction of these defects is imperative to achieve superconductivity in the bulk case, aligning with the requirements described in the previous paragraph.

3. In addition to the aforementioned issues, the incomplete removal of apical oxygen in Nd_{0.8}Sr_{0.2}NiO₂, as depicted in Fig. 5, introduces external structural distortion that impacts the flatness of the NiO₂ plane. This incomplete removal is considered a contributing factor to the absence of superconducting properties. The presence of residual oxygen atoms in the Nd/Sr plane disrupts the long-range order of the infinite-layer structure, leading to strong scattering and suppressing the superconductivity characterized by a low transition temperature, as discussed in a prior study on thin films [*Appl. Phys. Lett.* 123, 182601, 2023.]. Moreover, the insufficient topotactic reduction observed in the bulk sample is likely attributed to its large volume compared to thin films, which typically have a thickness of tens of nanometers. As a result, it is crucial to engineer the process of topotactic reduction to optimize the infinite layer and ultimately induce superconducting.

Based on these arguments, revisions have been made point by point in the following to further improve the manuscript.

*Many (nearly all) of the SC thin film samples also have a high (in fact, often higher) density of similar stacking faults, yet still exhibit clear signatures of SC. The presence of non-superconducting layers or inclusions also does not necessarily prevent superconductivity in this kind of layered sample. In principle, if there are regions which are sufficiently IL-like, then inclusions of secondary T' phases with the incorrect Ni filling could presumably just exist as passive non-superconducting layers, fundamentally no different from the inert substrate or capping layers in thin film samples. Similar effects are demonstrated in other layered oxide superconductors, eg. Sr₂RuO₄ (see *APL Materials* 10:041114, 2022).*

Response:

Thank you for the reviewer's comments and for providing insightful context from the literatures. We agree with the reviewer that many superconducting thin films exhibit a high density of similar defects to those found in bulk materials. However, these defects, such as non-superconducting layers or inclusions, are typically confined to the upper regions of the thin films. In contrast, the

superconducting layer with the infinite layer structure and proper hole doping near the substrate tend to be well-preserved throughout the sample or within a large area. Consequently, the signature of superconductivity is present in the electrical measurements.

In the case of bulk materials, the infinite layer phases are indeed present, but the continuity of the NiO₂ planes within the grains is disrupted by numerous and randomly distributed stack faults and interleaved structures. Furthermore, these striped infinite-layer phases even undergo orientation flips within the grain, as illustrated in Supplementary Fig. S13. Establishing a superconducting channel becomes challenging under these conditions. Therefore, eliminating or reducing these defects is necessary for achieving superconductivity in the bulk case.

In addition to the aforementioned issues, the incomplete removal of apical oxygen in Nd_{0.8}Sr_{0.2}NiO₂, as depicted in Fig. 5, introduces external structural distortion that impacts the flatness of the NiO₂ plane. This incomplete removal is considered a contributing factor to the absence of superconducting properties. The presence of residual oxygen atoms in the Nd/Sr plane disrupts the long-range order of the infinite-layer structure, leading to strong scattering and suppressing the superconductivity characterized by a low transition temperature, as discussed in a prior study on thin films [*Appl. Phys. Lett.* 123, 182601, 2023.]. Moreover, the insufficient topotactic reduction observed in the bulk sample is likely attributed to its large volume compared to thin films, which typically have a thickness of tens of nanometers. As a result, it is crucial to engineer the process of topotactic reduction to optimize the infinite layer and ultimately induce superconducting.

Revision:

These discussions have been incorporated into the Discussion part,

“The primary condition for the manifestation of superconductivity in nickelates is their metallic character, which is not observed in the bulk sample displaying insulating behavior. This may be induced by two main factors. Firstly, numerous stripe phases, such as the T'-type phase ($R_{n+1}Ni_nO_{2n+2}$), are present within the grains exhibiting insulating behavior³⁰. These phases originate from the parent RP phase of the Nd_{0.8}Sr_{0.2}NiO₃ compound before undergoing topotactic reduction, as shown in Supplementary Fig. S11 and S12. Then, these RP phases were transformed into the T'-type phase during the subsequent topotactic reduction process. Although the Nd₆Ni₅O₁₂ compound with a layer number of $n = 5$ exhibits superconductivity through self-doping³⁰, the layer number varies from place to place in the grain of bulk samples. Secondly, proper doping is essential for conferring metallic character and, consequently, inducing superconductivity. As discussed previously^{12-14,31}, the infinite-layer phase with a low hole doping level does not display superconductivity. Hence, despite the existence of a portion of the infinite-layer phase in the bulk sample with a certain amount of Sr, while the doping level may be lower compared to the threshold required for the superconducting phase. This can be corroborated by the observation of the segregation of Sr across the interface of the T'-type phase, resulting in a low Sr content in the infinite layer, as shown in Fig. 3. The lower concentration of Sr doping is also revealed by the reduced intensity of pre-peak of the O K edge in Supplementary Fig. S7, consistent with previous results³¹. Consequently, achieving superconductivity in bulk nickelates necessitates the preparation of a high-quality Nd_{0.8}Sr_{0.2}NiO₃ phase as the initial step. In cases where the RP phase cannot be completely eliminated, a higher concentration of Sr doping might be necessary to prevent underdoping caused by Sr segregation in the reduced RP phase (i.e., T'-type phase).

Also, it's important to note that many thin films, i.e. nickelates^{17, 20} and Sr_2RuO_4 ⁵⁵, exhibit superconductivity even in the presence of a high density of defects similar to those found in bulk materials. However, these defects, such as non-superconducting layers or inclusions, are usually confined to the upper regions of the thin films²⁰. In contrast, the superconducting layer with the infinite layer structure and proper hole doping near the substrate tend to be well-preserved throughout the sample or within a large area. Consequently, the signature of superconductivity is evident in electrical measurements. In the case of bulk materials, the infinite layer phases are indeed present, but the continuity of the NiO_2 planes within the grains is disrupted by numerous and randomly distributed stack faults and interleaved structures. Furthermore, these striped infinite-layer phases even undergo orientation flips within the grain, as illustrated in Supplementary Fig. S13. Establishing a superconducting channel becomes challenging under these conditions. Therefore, eliminating or reducing these defects is necessary for achieving superconductivity in the bulk case.

At last, the incomplete removal of apical oxygen in $\text{Nd}_{0.8}\text{Sr}_{0.2}\text{NiO}_2$, as depicted in Fig. 5, introduces external structural distortion that impacts the flatness of the NiO_2 plane. This makes the buckling of the NiO_2 planes, leading to enhanced scattering of electrons and inhibiting the pairing. Thus, this incomplete removal is considered to be the alternative possibility for the absence of superconductivity in the present bulk samples. The presence of residual oxygen atoms in the Nd/Sr plane disrupts the long-range order of the infinite-layer structure, leading to suppressed superconductivity characterized by a low transition temperature, as discussed in a prior study on thin films⁵⁶. Moreover, the insufficient topotactic reduction observed in the bulk sample is likely attributed to its large volume compared to thin films, which typically have a thickness of tens of nanometers. As a result, it is crucial to engineer the process of topotactic reduction to optimize the infinite layer and ultimately induce superconducting.”

Supplementary Fig. S13 | Discontinuity of NiO_2 planes within the grains. a and b The red and yellow dotted boxes correspond to domains with the orientations of $[100]$ and $[001]$, respectively.

Rather, the segregation of Sr near fluorite layers may indicate that the effective doping within IL

regions is too low for superconductivity. If the O-K and Ni-L edge EELS of the IL regions are consistent with other SC samples, then this does not yet explain the lack of observed SC.

Response:

Thanks for the reviewer's suggestion. We agree with the reviewer that the segregation of Sr near fluorite layers may result in lower doping within the infinite-layer regions, potentially responsible for the absence of superconductivity. To further explore this possibility, we compare the EELS results of the infinite-layer phases with those in superconducting infinite-layer nickelate thin films, which are nicely and systematically measured in a prior study [*Proc. Natl Acad. Sci. USA* 118, e2007683118, 2021.]. As shown in Supplementary Fig. S7 below, the pre-peak of the O K edge, indicative of the doping level [*Proc. Natl Acad. Sci. USA* 118, e2007683118, 2021.], is weaker than that in the superconducting thin films. This observation aligns with the underdoped concentration of Sr in the infinite-layer phase due to Sr segregation near fluorite layers. The shift of Ni $L_{3,2}$ edges towards lower energy is also consistent with the lower Sr doping, despite the broadening of the peaks, which might be attributed to the relatively lower EELS energy resolution in our experiments (~ 1 eV). Therefore, the lower level of Sr doping is likely responsible for the non-superconducting characteristics in the bulk sample.

Revision:

These discussions have been addressed in the revised manuscript as follows, Page 11, Line 225-235,

“The segregation of Sr near fluorite layers may result in lower doping within the infinite-layer regions, potentially responsible for the absence of superconductivity. To explore this possibility, we compare the EELS results of the infinite-layer phases with those in superconducting infinite-layer nickelates thin films, which are nicely and systematically measured in a prior study³¹. As shown in Supplementary Fig. S7, the pre-peak of the O K edge, indicative of the doping level³¹, is weaker than that in the superconducting thin films. This observation aligns with the underdoped concentration of Sr in the infinite-layer phase due to Sr segregation near fluorite layers. The shift of Ni $L_{3,2}$ edges towards lower energy is also consistent with the lower Sr doping, despite the broadening of the peaks, which might be attributed to the relatively lower EELS energy resolution in our experiments (~ 1 eV).”

Supplementary Fig. S7 | The EELS data of $\text{Nd}_{0.8}\text{Sr}_{0.2}\text{NiO}_2$ in this manuscript compared with those of Goodge, B. H [*Proc. Natl Acad. Sci. USA* 118, e2007683118, 2021.]. The weak pre-peak of O K edge corresponds to the underdoped level of Sr, as indicated by the red arrows. The shift of

Ni $L_{3,2}$ edges towards lower energy is consistent with the lower Sr doping, despite the broadening of the peaks, which might be attributed to the relatively lower EELS energy resolution in our experiments (~ 1 eV).

Comment 2: *Although their observation of chemical segregation and local distortions may be useful to other nickelate groups, their connection to superconductivity feels at this point a bit speculative. I would think that significantly more concrete insights could be gleaned at the mesoscale, for instance by considering the distribution of crystalline orientation at the tens or hundreds of nm length scales, e.g. across the fields of view shown in Figure 1 c-e. For instance, the authors point out “blurring” of the peaks in their SAED patterns, but these could also possibly be interpreted as bi-directional splitting of the peaks if the scattering area contained two domains with orthogonally oriented c-axes, the possibility of which is not discussed here. In fact, the peaks in figure 1e suggest either twin domains or a mostly unreduced phase: while strong anisotropy is seen in 1d between the 001 and 010 peaks, the pattern in 1e shows nearly isotropic lattice vectors in the 001 and 010 peaks. This distinction and a distribution of such twins could be clearly mapped out with scanning diffraction measurements and would be very useful to the field. Alternatively, atomic-scale characterization in different sub-regions of area 1e (i.e., across only horizontal or only vertical stripe regions (green boxes in attached) with comparisons of the local lattice orientation would also be useful.*

Response 2:

Thanks for the reviewer’s comments. We have addressed the issues you mentioned point by point to further improve the manuscript.

I would think that significantly more concrete insights could be gleaned at the mesoscale, for instance by considering the distribution of crystalline orientation at the tens or hundreds of nm length scales, e.g. across the fields of view shown in Figure 1 c-e.

Response:

We have conducted measurements of crystalline orientation using Transmission Kikuchi Diffraction (TKD) for the TEM sample. The Supplementary Fig. S4 provides an overview of the sample, capturing multiple grains and presenting the orientation distribution map. The results indicate a random orientation without any discernible textures within this mesoscale region. Therefore, the polycrystalline nature, coupled with the presence of numerous grain boundaries, is expected to increase resistance and deteriorate the insulating properties of the material, in contrast to single crystal thin films. These newly obtained experimental results have been incorporated into the revised manuscript.

Revision:

The following statements have been involved in the revised manuscript,

Page 7, Line 136-142,

“To gain insight into the grain orientation at the mesoscale, the crystalline orientation was determined using Transmission Kikuchi Diffraction (TKD) for the TEM sample (Methods and Supplementary Fig. S4). The results showed a random orientation without any discernible textures. Consequently, the polycrystalline nature, in conjunction with the abundance of grain boundaries, is

expected to increase resistance and degrade the insulating properties of the material, in contrast to single crystal thin films.”

Details of TKD in the Methods section:

“TKD experiments were conducted using TEM samples within the FIB-SEM. The sample was positioned on a designated sample holder to ensure that the electron beam was penetrating perpendicularly to the sample surface. The acquisition conditions of the electron beam were 30 kV, 30 nA, and the step size was 5 nm.”

Supplementary Fig. S4 | Crystalline orientation measured by Transmission Kikuchi Diffraction (TKD). a The BF image of Nd_{0.8}Sr_{0.2}NiO₂ including multiple grains. **b** The measured crystalline orientation map corresponding to the image in **a**.

For instance, the authors point out “blurring” of the peaks in their SAED patterns, but these could also possibly be interpreted as bi-directional splitting of the peaks if the scattering area contained two domains with orthogonally oriented c-axes, the possibility of which is not discussed here. In fact, the peaks in figure 1e suggest either twin domains or a mostly unreduced phase: while strong anisotropy is seen in 1d between the 001 and 010 peaks, the pattern in 1e shows nearly isotropic lattice vectors in the 001 and 010 peaks. This distinction and a distribution of such twins could be clearly mapped out with scanning diffraction measurements and would be very useful to the field.

Response:

Thank you for the reviewer's comments and suggestions. We apologize for any misleading statements in the manuscript. In fact, Figure 1e comprises two infinite layer structures with orthogonally oriented *c*-axes. An enlarged image of Selected Area Electron Diffraction (SAED) is presented in Supplementary Fig. S1b, where two sets of lattice vectors are indexed. To provide further evidence, scanning diffraction measurements based on the 4D-STEM technique, employing an Electron Microscope Pixel Array Detector (EMPAD) were applied to the interleaved region. Supplementary Fig. S2a and S2c display two typical diffraction patterns extracted from the orthogonal regions in the reduced grain, each featuring a *c*-axis orientation that is orthogonal to the other. Simultaneously, virtual imaging was reconstructed in Supplementary Fig. S2b and S2d by selecting the 011 spots, respectively. These results affirm that the area in Figure 1e contains two domains with orthogonally oriented *c*-axis. We have incorporated these experimental results and

discussions into the revised manuscript.

Supplementary Fig. S1 | **a** and **b** Enlarged views of selected area electron diffraction (SAED) in **Fig. 1e** (left) and **1f** (right), respectively.

Supplementary Fig. S2 | Analysis of the interleaved regions by Scanning diffraction measurements based on the four-dimensional STEM (4D-STEM) technique. **a** and **c** The typical nano-beam electron diffraction patterns extracted from the two orthogonal regions in the reduced grain. **b** and **d** Virtual dark-field images reconstructed by selecting the 011 spots, respectively.

Revision:

The following statements have been involved in the revised manuscript,

Page 6, Line 120-136,

“The high-angle annular dark-field scanning transmission electron microscopy (HAADF-STEM) image magnified in Fig. 1d reveals the presence of numerous stripes within the $\text{Nd}_{0.8}\text{Sr}_{0.2}\text{NiO}_2$ grains. Further examination of the interior grains provides clear stripe contrast in Fig. 1e and 1f. Some of these stripes align parallel to the [010] direction (highlighted by a yellow box in Fig. 1d), with the elongated diffraction spots extending along the [001] direction in the inset of Fig. 1e. For regions displaying a vertically staggered distribution (marked by a red box in Fig. 1d and Fig. 1f), the bi-directional splitting of the diffraction spots indicates that the area contains two domains with orthogonally oriented c -axis, as marked by the lattice vectors in the inset of Fig. 1f. The enlarged images of Selected Area Electron Diffraction (SAED) are presented in Supplementary Fig. S1, where two sets of lattice vectors are indexed. Additionally, scanning diffraction measurements based on the four-dimensional STEM (4D-STEM) technique, utilizing a hybrid pixelated detector, were applied to the interleaved region. The typical diffraction patterns extracted from the two orthogonal regions provide evidence of domains with orthogonal orientation, as further supported by the virtual 4D-STEM imaging (Supplementary Fig. S2) and atomic resolution images of sub-regions within the interleaved area (Supplementary Fig. S3).”

The experimental details of scanning diffraction measurements based on the 4D-STEM technique have been involved in the Methods section as follows,

“4D-STEM NBED data were acquired using a small convergence semi-angle 0.3 mrad, a camera length of 1.45 m, and an Electron Microscope Pixel Array Detector (EMPAD), the customized python scripts were utilized to select specific diffraction points for virtual imaging.”

Alternatively, atomic-scale characterization in different sub-regions of area 1e (i.e., across only horizontal or only vertical stripe regions (green boxes in attached) with comparisons of the local lattice orientation would also be useful.

Response:

As suggested by the reviewer, atomic resolution images of sub-regions within the interleaved area are presented in Supplementary Fig. S3. The crystal orientations are indexed in Supplementary Fig. S3b and S3c, revealing that the horizontal and vertical stripe domains in this grain consist of infinite layer phases but are orthogonal to each other. Supplementary Fig. S3 has been incorporated into the revised manuscript.

Supplementary Fig. S3 | Atomic resolution images of sub-regions within the interleaved area.

a The HAADF image of the same grain as **Fig. 1f**. **b** and **c** The enlargement of the corresponding color box areas in **a** respectively.

In addition to the above suggestion to significantly improve the scope and impact of their work, I have some technical questions and general comments below:

Response:

Thanks for the reviewer's comments. We have revised the above issues you mentioned point by point to further improve the manuscript.

Technical questions:

Question 1. *What are the characteristic domain sizes of the precursor as compared to the reduced crystals?*

Response 1:

Thanks for the reviewer's comment. We conducted new experiments to characterize the structure of the $\text{Nd}_{0.8}\text{Sr}_{0.2}\text{NiO}_3$ compound before topotactic reduction, as illustrated in Supplementary Fig. S11 and Fig. S12 below. The nickelate parent phase inherently harbors numerous defective structures, including the RP phase. This cross-arrangement of defective structures constitutes a significant factor contributing to the formation of three-dimensional block-like structures. The characteristic domain size of the precursor $\text{Nd}_{0.8}\text{Sr}_{0.2}\text{NiO}_3$ ranges from tens to hundreds of nanometers in length and several to tens of nanometers in width, as illustrated in Supplementary Figure S11 for the sample before reduction. This size is comparable to that observed in the reduced sample. These results have been incorporated into the Supplementary Information.

Revision 1:

The following discussions have been added in the Supplementary Information,

"In STEM characterization of bulk $\text{Nd}_{0.8}\text{Sr}_{0.2}\text{NiO}_3$ before topotactic reduction, inherent defect structures similar to the RP phase were observed in the grains of its parent phase (Supplementary Fig. S11). The characteristic domain size of the precursor $\text{Nd}_{0.8}\text{Sr}_{0.2}\text{NiO}_3$ ranges from tens to hundreds of nanometers in length and several to tens of nanometers in width. These defects likely contribute to the formation of a 3D block-like structure within grains after reduction. The results of

EDS mapping showed that the distribution of elements was uniform as that of the reduced samples, as shown in Supplementary Fig. S11a. This size of grains is comparable to that observed in the reduced samples. The magnifying bright-field (BF) image in Fig. S11b and Fig. S11d reveals the presence of numerous stripes within the $\text{Nd}_{0.8}\text{Sr}_{0.2}\text{NiO}_3$ grains, similar to the block-like structure in the reduced samples. The block-like configuration within the parent phase consists of alternating RP phases, each with a distinct c -axis orientation, as illustrated in Supplementary Fig. S12. Consequently, the defect structures present in these parent phases serve as the origin of the block-like structure observed in the $\text{Nd}_{0.8}\text{Sr}_{0.2}\text{NiO}_2$ grains after following topotactic reduction.”

Supplementary Fig. S11 | Structure characterization of bulk $\text{Nd}_{0.8}\text{Sr}_{0.2}\text{NiO}_3$ before topotactic reduction. **a** EDS mapping of $\text{Nd}_{0.8}\text{Sr}_{0.2}\text{NiO}_3$ polycrystalline at low magnification. **b** and **d** High-resolution BF-STEM image of $\text{Nd}_{0.8}\text{Sr}_{0.2}\text{NiO}_3$. **c** and **e** The magnified HAADF-STEM images of marked regions in **b** and **d**.

Supplementary Fig. S12 | Atomic structure in bulk $\text{Nd}_{0.8}\text{Sr}_{0.2}\text{NiO}_3$ before topotactic reduction.

a Atom model of RP phase ($n = 4$). **b** Atomic-resolved HAADF-STEM image of **Fig. S11c**. **c** and **d** The ϵ_{xx} and ϵ_{yy} strain maps are calculated by GPA for **b**. **e** Atom model of RP phase with c -axis along electron beam direction (the oxygen atoms are not shown here). **f** Atomic-resolved HAADF-STEM image of **Fig. S11e**. **g** and **h** The ϵ_{xx} and ϵ_{yy} strain maps are calculated by GPA for **f**.

Question 2. Lines 170-172: *O-K edge EELS of the IL and T' phases: where precisely are the different spectra collected from? Do these regions include summing over the fluorite layers, or only the regions between them?*

Response 2:

Thanks for the reviewer's comment. To improve the signal-to-noise ratio, we integrated the signal over the regions as indicated by the red boxes below. The signal for IL is extracted from the pure IL region away from the T'-type phase, while the signal for the T'-type phase includes the fluorite layer and the nearby IL region, as marked by the red boxes in Supplementary Fig. S6a. This information has been involved in the revised manuscript.

Supplementary Fig. S6 | EELS measurements. **a** EELS acquisition at different locations in **Fig. 3c**. To improve the signal-to-noise ratio, integrate the signal over the regions as indicated by the red boxes. **b** EELS mapping at atomic resolution of the T'-type phases.

Revision 2:

We have added the following statements in the revised manuscript,

Page 11, Line 214-215,

“To improve the signal-to-noise ratio, the EELS signals in Fig. 3c are integrated over the regions as marked in Supplementary Fig. S6a.”

Question 3. *The pre-edge region in the O-K EEL spectrum indicates a poor background fit/subtraction. Can the authors improve this to obtain a flat, zero-values pre-edge?*

Response 3:

Thanks for the reviewer's comment. We have updated Fig. 3c (as shown below) by re-subtracting the background to obtain a flat pre-peak in the revised manuscript.

Fig. 3 | EDS and EELS measurements of $\text{Nd}_{0.8}\text{Sr}_{0.2}\text{NiO}_2$. **a** EDS elemental mappings of the stripe structure in $\text{Nd}_{0.8}\text{Sr}_{0.2}\text{NiO}_2$. **b** The intensity profile of elemental distributions in **a**. **c** EELS O K, Ni L, and Nd M of infinite-layer phase (black) and T'-type phase (red).

General:

Overall, I found the results of the paper presented clearly, but the introduction is not very coherent and should be revised for clarity and accuracy. There are several instances of the authors trying to tie together unrelated conclusions and studies which is a bit confusing, for example:

Question 4. *Paragraph 1 seems mostly about the d electron filling, but also tries to bring in the dimensionality and cites two papers which are only secondarily concerned with the formal d filling but rather more with the parent Mott-Hubbard electronic landscape (Hepting) and its evolution with added holes (Goodge). The trilayer sample (Li) doesn't fall within the superconducting dome. These are all important references certainly worth including, but the relation of each to its discussion in the paper could be improved.*

Response 4:

Thanks for the reviewer's comment. We are sorry for these misleading statements in the manuscript. We have rewritten the **Introduction** section in the revised manuscript with appropriate references (see the revised **Introduction** below).

Question 5. *Line 59 refers to "different ionic radii" as the reason for RP faults in thin films. Actually there are two distinct challenges being conflated here: the RP faults arise mostly because of the high tensile strain in precursor films grown on substrates which are optimized for the IL phase (doping has some subtle contribution here, but far more important is the primary R radius, e.g. La vs Nd (see APL Materials 8:4, 041107, 2020; Advanced Materials, 2104083, 2021; Nature Communications 14:1468, 2023). The extra difficulty for doped films relates also in large part to the high Ni valence which must be adopted in the doped precursor (see Sci. Adv. 9, eadh3327, 2023).*

Response 5:

Thanks for the reviewer's useful suggestions and comments. We agree with the reviewer that the main difficulties in the synthesis of the sample precursor are the high tensile strain in precursor films grown on substrates and the instability of the high-valence state Ni. The radius of the doped ion has little influence on the structure compared with the radius of the primary R . We have modified this part of the revised manuscript (see the revised **Introduction** below).

Question 6. *Discussion of various RP phases: the RP defects which form in precursor films under tensile strain are distinct from the layered RP compounds which are intentionally grown to stabilize intermediate nickel valences. The bilayer La compounds under pressure are so far understood to be quite different from the reduced layered compounds.*

Response 6:

Thanks for the reviewer's suggestions. We note that these two different RP phases, one is that the RP phase is a stoichiometric structure of intentional growth, and the other is a defect phase above the infinite layer film. For the sake of distinction, we name them the layered RP phase and the RP defects, respectively. For layered RP phase, i.e., $\text{La}_3\text{Ni}_2\text{O}_7$ single crystals, the superconducting mechanism of metallization of interlayer Ni-O σ bonds under high pressure is very different from that of infinite layer nickelates. This information has been emphasized in the revised **Introduction**.

Revision:

We have rewritten the Introduction in the revised manuscript as follows,

“The emergence of superconductivity in nickelates represents a pivotal breakthrough in the exploration of superconducting materials and mechanisms¹, which is underscored by the notable similarity in the atomic configuration and electronic characteristics between the infinite NiO_2 layers in nickelates and the CuO_2 planes in cuprates²⁻⁶. The initial discovery occurred in $\text{Nd}_{0.8}\text{Sr}_{0.2}\text{NiO}_2$, achieved through the topotactic chemical reduction of the perovskite precursor $\text{Nd}_{0.8}\text{Sr}_{0.2}\text{NiO}_3$ using CaH_2 ⁷⁻¹¹. Subsequently, other nickelates thin films with diverse compositions, such as $R_{1-x}\text{A}_x\text{NiO}_2$ ($R = \text{La, Pr, Nd, and A} = \text{Sr, Ca, Eu}$), have demonstrated superconductivity under appropriate doping concentrations¹²⁻¹⁹.

Two crucial factors contributing to the observed superconductivity are the infinite layer structure and proper hole doping achieved through the alkaline earth substitution²⁰. The infinite-layer structure is typically derived from the precursor perovskite phase. However, synthesizing a pure perovskite phase, i.e., $\text{Nd}_{0.8}\text{Sr}_{0.2}\text{NiO}_3$, remains challenging due to structural instability²⁰. This instability is primarily attributed to the high tensile strain in precursor films grown on substrates¹⁹⁻²¹ and the instability of the high-valence state Ni^{18,22}. Consequently, the Ruddlesden-Popper phase (RP, $R_{n+1}\text{Ni}_n\text{O}_{3n+1}$, R is rare-earth) is often observed in nickelates thin films, hindering the manifestation of superconductivity^{23,24}. Recent developments even show that intentionally grown layered RP phase $\text{La}_3\text{Ni}_2\text{O}_7$ can exhibit superconductivity with a T_c of 80 K under pressure²⁵. However, this superconducting mechanism differs from that of infinite-layer nickelates by stabilizing intermediate nickel valences^{25, 26}. These orthorhombic RP phases ($R_{n+1}\text{Ni}_n\text{O}_{3n+1}$) typically transform into the tetragonal T' -type phase ($R_{n+1}\text{Ni}_n\text{O}_{2n+2}$) through oxygen deintercalation and structural rearrangement after topotactic reduction^{10,27}. The T' -type phase possesses quasi-two-dimensional NiO_2 planes in square-planar coordination with rare-earth fluorite blocking slabs interleaved in every n nickel layer^{7, 28-30}. Yet, only a specific case, $\text{Nd}_6\text{Ni}_5\text{O}_{12}$ with $n = 5$, has been

found to exhibit superconductivity with outer electrons $d^{8.8}$ ³⁰. These intricate phenomena manifest the strong correlation between the local atomic structure and superconductivity in nickelates. For the hole doping mechanism, the electronic structure measurements have provided insights within the Mott-Hubbard regime in the parent phase NdNiO₂⁶. The doping evolution has been convincingly demonstrated through locally resolved electron energy-loss spectroscopy (EELS)³¹. The interfacial reconstruction has been revealed to stabilize the external crystal structure and alleviate polar discontinuity in Nd_{0.8}Sr_{0.2}NiO₂ thin films³²⁻³⁵. Despite significant strides in addressing these key issues and advancing our understanding of superconductivity in nickelates thin films, the reasons for the absent superconductivity in bulk nickelates remain elusive³⁶⁻³⁸.”

Minor comments:

Question 7. *In the abstract line 35: I think the “even” should be removed: by definition, a fully reduced film will have no residual apical oxygens, so the comment refers to the (very common) issue of partially incomplete reduction.*

Response 7:

Thanks for the reviewer’s comment. We have deleted “even” in line 35 of the revised manuscript.

Question 8. *In line 52: “confirmed” seems strong given H-free reduction of superconducting samples (see Phys. Rev. Materials 7, 013802, 2023) and the lack of confirmation by other groups of the H doping effect. I would recommend replacing with “suggested”, “proposed”, or “explored”. In fact, relating to my previous comments, it could simply be removed as the importance or not of H does not seem related to the conclusion of quasi-2D-like structure (which is anyways implied by the crystal structure regardless of H impurities).*

Response 8:

Thanks for the reviewer’s suggestion. We agree with it that there is still a lack the confirmation of the H doping effect by other groups. We have removed the expression in the revised manuscript.

Question 9. *In line 122: “additional oxygen atoms” in reduced RP fluorite layers are not actually additional, these are the same O which were already present in the rock salt, but the configuration has changed. The compound has overall fewer O than in the precursor phase.*

Response 9:

Thanks for the reviewer’s comment. We apologize for the confusion caused by the wording in the manuscript. As the reviewer said, the oxygen layer is formed through the rearrangement of oxygen atoms during the reduction process rather than being inserted.

To prevent any potential misinterpretation, the sentence “Moreover, the pristine rock salt layers are changed into the fluorite structure after topotactic reduction, with additional oxygen atoms inserted between two *R* layers as indicated by the red spheres in Fig. 2a.” has been revised as “Furthermore, the rock salt layers undergo a transformation into the fluorite structure after topotactic reduction, involving the rearrangement of oxygen atoms between two *R* layers, as illustrated by the red spheres in Fig. 2a.”.

Reviewer #3:

Comment: *The paper discusses how microstructure and local atomic arrangements in Nd_{0.8}Sr_{0.2}NiO₂ can lead to a non-superconducting state. These atomic scale observations range from point defects of F to remaining (non-reduced) oxygen sites. The authors do a good job relating these observations back to prior works, building upon a preexisting framework of knowledge and leave plenty for future researchers to follow up on. The techniques are adequate and appropriate for the purpose of the current paper. I quite enjoy the variety of techniques used, ranging from classic diffraction to newer imaging, to high resolution EELS. The figures are altogether attractive, and the subject of the manuscript is a “hot topic”, so Nature Communication is an appropriate journal. There are a few points that should be addressed to strengthen the validity of certain claims. I have also provided a few suggestions for modified analysis that would provide more reliable metrics and statistics.*

Response:

Thanks very much for the reviewer’s positive comments about our attractive figures. We very much agree with your proposed suggestions. The revisions have been made point by point in the following to further improve the manuscript.

Question 1. *In Figure 1, the streaks mentioned in the diffraction pattern are not visible. Zoom in to show the first (an maybe second) order Bragg peaks to make things more clear. Add the full diffraction as supporting if you like.*

Response 1:

Thanks for the reviewer’s suggestion. For improved clarity, we have annotated the diffraction patterns of Fig. 1 and zoomed in to show the second order Bragg peaks. Also, the enlarged views are incorporated into Supplementary Fig. S1.

Supplementary Fig. S1 | **a** and **b** Enlarged views of selected area electron diffraction (SAED) in Fig. 1e (left) and 1f (right), respectively.

Revision 1:

We have added the following statements in the revised manuscript,

Page 6, Line 129-130,

“The enlarged images of Selected Area Electron Diffraction (SAED) are presented in Supplementary Fig. S1, where two sets of lattice vectors are indexed.”

Question 2. *In Figure 2, I assume that ϵ_{yy} is the [001] axis. This should be clarified. I also encourage the author to not use GPA. They have beautiful images from which the atom positions can be directly extracted. From the positions they can map average bond lengths in each row directly and provide error bars. GPA can be subject to artifacts. The same should be done for Fig 4.*

GPA artifacts: <https://doi.org/10.1016/j.ultramic.2015.05.020>

iDPC/HR-STEM profiles with error bars:

<https://doi.org/10.1038/s41586-021-04238-z>,

<https://doi.org/10.1002/adma.202207736>,

<https://doi.org/10.1016/J.ULTRAMIC.2017.06.002>

Response 2:

Thanks for the reviewer’s useful suggestions. The ϵ_{yy} is indeed the strain distribution along the [001] direction in Fig. 2. Following your suggestion, we have incorporated the measured bond length in Fig. 2. The analysis of bond lengths in Fig. 4a has been added in Supplementary Fig. S8. Nevertheless, we have retained the GPA distribution to assist readers in clearly comprehending the inhomogeneous strain distribution within the grain caused by these defect layers. The related references as mentioned by this reviewer have been approximately cited as well.

Revision 2:

We have added the following statements in the revised manuscript,

Page 8, Line 172,

“The decrease in lattice spacing within the fluorite layers is clearly demonstrated by the bond lengths depicted in Fig. 2e. This finding is consistent with the outcomes of the geometrical phase analysis (GPA) presented in Fig. 2f, where the compressive strain ϵ_{yy} attains a negative peak at the fluorite layers.”

Page 13, Line 252,

“The vertical and parallel stripes are distinctly discernible through GPA in Fig. 4c and 4d. To address potential artifacts associated with GPA⁴⁵, we simultaneously conducted measurements of bond lengths, as previously undertaken⁴⁶⁻⁴⁸. The variations in lattice spacing within these stripes are also clearly resolved, as illustrated in Fig. S8.”

Figure 2 has been updated in the revised manuscript as shown below.

Fig. 2 | Atomic structure of the stripe domains in $\text{Nd}_{0.8}\text{Sr}_{0.2}\text{NiO}_2$. **a** Atom model of layered T'-type $(\text{Nd}_{0.8}\text{Sr}_{0.2})_4\text{Ni}_3\text{O}_8$ ($n = 3$). **b** Atomic-resolved iDPC-STEM image of the T'-type phase along [100] direction with the light elements in infinite and fluorite layers clearly resolved. The atomic model is overlaid to guide the eyes. **c** Atomic-resolved HAADF-STEM image of the stripes under [100] orientation, and the atomic models without oxygen are overlaid accordingly. To differentiate between T'-type phase and infinite layer structures, the latter are simply defined here as $n > 10$ layers. **d** The intensity profile across adjacent Nd atomic columns is extracted along the direction indicated by the green arrow in **c**. The orange arrows are used to mark the fluorite layers. **e** Averaged bond lengths between R-site atoms along [001] direction. **f** Strain ϵ_{yy} along the [001] calculated by GPA for **c**.

Supplementary Figure S8 has been added in the Supporting Information as shown below.

Supplementary Fig. S8 | Lattice parameter analysis for the stripe structure. a Atomic resolution HAADF-STEM images of the parallel and vertical stripes in **Fig. 4a. b** Averaged bond lengths of the parallel and vertical stripes measured from the red dotted box in **a**.

Question 3. *In Figure 3, are the EDS atomic resolution? It does not appear so. The data is good, but stating that these are high-resolution is an overreach.*

Response 3:

Thanks for the reviewer's comment. The statement has been corrected in the revised manuscript.

Revision 3:

The following statements have been altered in the revised manuscript, Page 10, Line 191,

The sentence "Furthermore, high resolution STEM-EDS measurements were carried out to analyze the element distributions in Fig. 3a and 3b." has been revised as "Furthermore, high-magnification STEM-EDS measurements were carried out to analyze the element distributions in Fig. 3a and 3b."

Question 4. *Regarding, "The larger ratio of L3/L2 corresponds to the higher valence of Ni". There is a common misconception that the L3/L2 ratio results from valence state changes. The ratio is highly correlated with valence state. The real underlying reason for the ratio is crystal field splitting*

that or electron correlation. Crystal field splitting results from changes in coordination, which is in turn related to valence state and why L_3/L_2 trends so strongly. It would be best to state the correct physics regarding the coordination change then make it clear that a change in valence state is inferred.

Response 4:

Thanks for the reviewer's comment. We have involved these statements in the revised manuscript.

Revision 4:

**The following statements have been changed in the revised manuscript,
Page 11, Line 221,**

The sentence "The larger ratio of L_3/L_2 corresponds to the higher valence of Ni." has been revised as "The larger ratio of L_3/L_2 corresponds to a higher valence of Ni, which is attributed to the alterations in crystal field splitting arising from changes in the coordination environment of Ni."

Question 5. *"Remarkably, there is still residual apical oxygen in Nd/Sr plane after topotactic reduction, as indicated by the orange arrows in Fig. 5c - 5e with a random distribution." Do you mean "Remarkably, there is still a random distribution of residual apical oxygen in Nd/Sr plane after topotactic reduction, as indicated by the orange arrows in Fig. 5c - 5e."*

Response 5:

Thanks for the reviewer's comment. We have corrected the statements in the revised manuscript, as suggested by the reviewer.

Revision 5:

**The following statements have been altered in the revised manuscript,
Page 15, Line 296,**

The sentence "Remarkably, there is still residual apical oxygen in Nd/Sr plane after topotactic reduction, as indicated by the orange arrows in Fig. 5c - 5e with a random distribution." has been revised as "Remarkably, there is still a random distribution of residual apical oxygen in Nd/Sr plane after topotactic reduction, as indicated by the orange arrows in Fig. 5c - 5e."

Question 6. *You make a strong point that the materials are sensitive to defects. Most nickelates are prone to damage from sample preparation and the electron beam. How have you determined that the FIB Ga-beam or the electron beam have not resulted in the observed structure and lack of atomic sites?*

Response 6:

Thanks for the reviewer's comment. The damage caused by the electron beam was examined in our experiments. The electron beam currents used for image acquisition and EELS measurements were 30 pA and 70 pA, respectively. For EDS measurements, the current was set at 300 pA for low magnification and 70 pA for high magnification. No obvious beam damage or atom displacement was evident during the HAADF and iDPC experiments. This was confirmed by capturing multiple images at the same location. Additionally, the sample was inspected both before and after EDS and

EELS measurements due to the high electron dose and total exposure time involved in these procedures. No significant changes in contrast were observed. The O-K edge, which is sensitive to the irradiation damage in oxides, was also examined before and after STEM-EELS acquisition, with no noticeable changes observed. Therefore, we conclude that electron beam damage does not exert a significant impact on the as-grown structures or introduce notable defects.

When preparing TEM samples using FIB, we employed a low voltage (2 kV) to minimize damage, particularly during the thinning and removal of the surface amorphous layer. Despite not observing Ga contamination in the sample through EDS results, we acknowledge that the absence of such detection does not definitively rule out potential defects or structural transitions induced by the Ga ion beam. Notably, in the previous literature, such as in publications like [*Nat. Commun.* 15, 378, 2024., *APL Mater.* 8, 041107, 2020., *Sci. Adv.* 7, eab18091, 2021.], where TEM characterization of bulk or thin films of nickelates involved FIB sample preparation, there was no explicit mention or reporting of significant structural transitions caused by the Ga ion beam. This information leads us to believe that the Ga ion beam might not play a substantial role in inducing observable effects. However, to further eliminate this potential effect, we try an alternative approach by grinding the sample, dispersing it into alcohol, and subsequently transferring it to the TEM grid. Note that we avoided using the polishing method due to the fragility of the reduced sample. As depicted in Supplementary Fig. S14, the observed stripe structures and defects remain consistent, providing additional evidence that rules out structural transitions or defects induced by FIB processing.

Revision:

In the Methods section, we have involved the following statements,

“The electron beam current utilized for HAADF-STEM and iDPC-STEM acquisitions was 30 pA. STEM-EELS experiments were conducted with a probe current of 70 pA, a collection angle of 100 mrad, a pixel time of 0.03 s, and a dispersion of 0.3 eV per channel. In the case of EDS measurements, the current was set at 300 pA for low magnification and 70 pA for high magnification. No apparent beam damage or atom displacement was observed during the HAADF and iDPC experiments, as confirmed by capturing multiple images at the same location. Furthermore, the sample was examined both before and after EDS and EELS measurements due to the substantial electron dose and total exposure time associated with these procedures. No significant changes in contrast were noted. The O-K edge, which is sensitive to the irradiation damage in oxides, was also examined before and after STEM-EELS acquisition, with no noticeable changes observed. Therefore, we conclude that electron beam damage does not have a significant impact on the as-grown structures or introduce noticeable defects.”

“The cross-sectional TEM lamellas of $\text{Nd}_{0.8}\text{Sr}_{0.2}\text{NiO}_2$ was thinned to electron beam transparency at 30kV by using Carl Zeiss Crossbeam 550L FIB-SEM, followed by the removal of the surface amorphous layer at 2 kV. In addition, to exclude potential structural transition or defects induced by the Ga ion beam during TEM sample preparation, we chose an alternative approach by grinding the sample, dispersing it into alcohol, and subsequently transferring it to the TEM grid. Note that we abstained from using the polishing method due to the fragility of the reduced sample. As depicted in Supplementary Fig. S14, the observed stripe structures and defects remain consistent, providing additional evidence that rules out structural transitions or defects induced by FIB processing.”

Supplementary Fig. S14 | TEM observations of samples by grinding. To rule out structural transitions or defects induced by FIB processing, we chose an alternative approach by grinding the sample, dispersing it into alcohol, and subsequently transferring it to the TEM grid. The BF images show the stripe structure and defects within $\text{Nd}_{0.8}\text{Sr}_{0.2}\text{NiO}_2$ grains.

REVIEWER COMMENTS

Reviewer #1 (Remarks to the Author):

In their revised manuscript, Kejun Hu et al. have introduced a host of text additions and changes together with many new microscopy figures, in response to the comments of all three reviewer. I find that these changes have improved the manuscript significantly, and congratulate them for their efforts. The comments and queries from the reviewers have been treated seriously and in depth and, I believe, are all successfully addressed. The readability and messaging are also now much improved. Therefore, I do not hesitate to recommend the publication of this work, which will make a nice article for Nature Communications. Nevertheless, to maximize the standard of the work, I do have a few minor points which should first be addressed, as now detailed.

Line 100-101, proposed edit: "While the parent phase Nd_{0.8}Sr_{0.2}NiO₃ has an orthorhombic structure with (*Pbnm*) lattice parameters of $a_0 = 5.39 \text{ \AA}$, $b_0 = 5.38 \text{ \AA}$, and $c_0 = 7.61 \text{ \AA}$, here we use the pseudocubic unit cell..."

Line 103, proposed edit: "... orientation relationship between..."

Line 104: would be more systematic to have (100)_{pc} // (1-10)_O
In my opinion, it would also be preferable to define the orientation relationships by lattice vectors not planes (i.e. swap the round brackets for square ones).

Line 110: I do not find this line clear. If resistance is elevated by grain boundaries and voids, the material should become more insulating. From the phrase "deteriorate the insulating properties", it is not clear if the authors mean that it becomes more or less insulating...

Line 113: similarly, perhaps it should be "despite the reduction in insulating properties"?

Line 115: more appropriate might be "... grain boundaries and voids is not the primary detrimental effect...?"

Line 122: "... examination of the grain interiors..."

Line 141-142: as above, "increase resistance and degrade the insulating properties" is confusing.

Fig. 1(a). Would be more obviously consistent with the right-handed rule – and more consistent with panels (e) and (f) – if [010]_{pc} points to the right. Or, the left-pointing arrow should be indexed as [0-10]_{pc}.

Line 160: perhaps better as "the rock salt layers of the RP phase"?

Line 296-297: perhaps better as "Remarkably, there are still randomly-distributed residual apical oxygen atoms in the Nd/Sr plane...?"

Line 356: the "while" seems unnecessary.

Line 366: "Also, it's important to note..." -> "It is important to note..."

Line 382: perhaps better as "This makes the NiO₂ planes buckle...?"

Supp. Fig. S2(c): indexing inconsistent with right-hand rule.

Supp. Fig. S13: indexing of directions in yellow regions inconsistent with right-hand rule. Note [001] must be coming towards the viewer, because [100] is coming towards the viewer in the red regions. Either point [100] in the opposite direction, or label as [-100]. (*Or* do likewise for the [010] arrow.)

Reviewer #2 (Remarks to the Author):

I appreciate the efforts by Hu, et al. to improve the introduction and discussion of their manuscript, and the additional experiments which are now included. I feel that the topic of different kinds of disorder which are characterized here are useful information to the community, but for this same reason feel it will be crucial that their presented results are as accurate and accessible as possible, especially to a broad audience not familiar with microscopy techniques. I therefore have a few additional suggestions which I feel will greatly improve the clarity of the article for publication in Nature Communications.

For the strengths of the article, I feel the title is not appropriate. It suggests a definitive or smoking gun solution to the "absence of superconductivity", while the results here simply offer insight to the many complicated effects which are likely all contributing in some way. I suggest something along the lines of "Atomic scale disorder and reconstruction in bulk infinite-layer nickelates" would be more appropriate.

I do not understand the argument in lines 113: "despite the improvement in insulating properties under high-pressure conditions, superconducting properties remained elusive. This suggests that the presence of grain boundaries and voids does not have a detrimental effect on superconductivity". Are the authors trying to argue that the intra-grain crystalline regions are themselves insulating? This is confusing to me, since other reports of bulk square-planar nickelate single crystals exhibit metallic transport (though still no superconductivity), see Li et al. Sci. Adv. 9, eade4418 (2023) 10.1126/sciadv.ade4418 or <https://arxiv.org/pdf/2403.00493.pdf>. If the authors argue that the voids and boundaries do not significantly influence the electronic transport, can they include a resistivity trace for the precursor powder? Presumably a high-quality precursor should also exhibit metallic behavior until at least 100-200 K (even lower if properly doped).

The EDX maps presented in Figure 3 appear highly aliased. My guess would be an aggressive Gaussian or Weiner filter applied by the EDX acquisition and processing software. I suggest the authors replace what is currently presented with the unfiltered data, or carefully choose a much lower filter and report it in the methods with unfiltered data in the Supplement. Likely also in Figure 1 and Supplemental Figure S11.

In figure 5, the color bars in panels g and i are confusing because they present continuum parameters on a diverging colormap. The figures would carry much more physical intuition if the authors remap the red-blue color scale so that the middle value (white) corresponds to a physically relevant "middle" value, eg. the average c/a ratio of 0.86 (or 0.886) and 180 degree bond angle.

Actually, I do not find that the bond angle mapping by iDPC adds much useful information at all to the rest of the manuscript, beyond what one could already expect from the many other kinds of disorder discussed. Furthermore, given that the Ni displacements shown in Figure 5i bond angle map all show a systematic "up" component and given the extreme sensitivity of iDPC to very small sample mistilt, thickness, and other projection artifacts, I wonder about the robustness of these results. In regard to the speculation that "it could therefore be inferred that the strain could suppress the local distortions and potentially flatten the NiO₂ planes", I do not understand why intuitively one would expect a square planar phase to have any kind of systematic bond distortion along one direction, which would impart the bulk compound with an overall mirror symmetry-breaking. To me, this part of the discussion feels far too speculative, and without significantly more effort to demonstrate its reproducibility I would be hesitant to include it here.

Minor comments:

Lines 69-70 "Yet, only a specific case, Nd₆Ni₅O₁₂ with n = 5, has been found to exhibit superconductivity with outer electrons d_{8.8}" – I suggest removing this comment, as results recently presented by the Mundy group suggest other layered phases can also host superconductivity (<https://meetings.aps.org/Meeting/MAR24/Session/G16.5>).

The legend reference in supplementary figure 7 to "Berit H. Goodge" seems like a bit of an odd way to reference published work – perhaps the authors can replace it with a simple citation

number or something like "Goodge, et al."

Reviewer #3 (Remarks to the Author):

The authors of the manuscript have done a fairly good job responding to the previous reviews. The manuscript is even better than the initial submission, which was also good. The new supporting figures are very nice. However, they did miss a couple of the key comments addressed in the first review. I think that as a whole the authors conclusions are correct, however there is a possibility of miss interpreted EELS data that can only be addressed with more robust data analysis and thought into physical origins, and not simple rewording of sentences. I suggest that the EELS be given more thought before publication. However, given that the different characterization techniques lead to the same conclusion and the conclusions are logical. I feel the manuscript if left as would still be of high quality and leave it to the editor's decision if the suggested EELS consideration are necessary.

More details:

Question 1: They included a new supporting figure where the streaks in the diffraction pattern are clearer, but they have not changed the main text figure as they stated in the response. The streaks in the diffraction patterns in Fig1 d and e are still not visible but are main discussion topics of the text.

Question 4:

The change could still be from correlation and not valence state change. It is in general difficult to understand the physical correlation to ELNES in such a complex system that has so many possible things effecting electronic fine structure. The Goodge, B. H. reference makes a point of this by showing that the Ni-L3 edge is sensitive to doping and oxygen content. The authors make a comment that the Sr-doping decreases the Ni-L3 edge energy, but according to the Goodge, B. H. reference the Ni-L3 edge should increase with Sr and is also sensitive to the oxygen content. The authors have mentioned that the valence state of Ni has changed based on the intensity but do not see any peaks shifts. In addition, the cited paper shows that the Ni-L3 peak energy and shape is sensitive to the oxygen content, while the Ni-L2 edge shape changes and the onset remains the same energy. Again, the authors see a change in the intensity ratio but no change in the peak energies, which is odd and could suggest that something else is resulting in fine structure changes. Although valence state change is most likely the cause for the different intensity ratio's other things could be happening and the other changes observed with valence state changes in other studies are not observed in this manuscript. The authors adjusted their wording accommodate my comment regarding changes in molecular orbital theory and crystal field splitting but did not dive into the other physics that might be resulting in fine structures changes.

Question 6:

It sounds like the authors were careful to manage the current during spectroscopy and evaluate possible beam damage. However, they have presented none of the evidence and this is a question that will be asked by any expert in the field who read the paper. The current is not the main issue when it comes to damage. The main factors are accumulated dose and operating energy. 300 kV is an energy that most oxides damage at, especially with the dose requirements to perform ELNES analysis as performed in the manuscript. I know that this was a consideration for Goodge, B. H. whom the authors cite. I believe that the authors have been thorough in their experiments, but I ask that the authors add in data regarding exposure to the beam.

RE: Manuscript Number: NCOMMS-23-54994A

“Atomic origin of absent superconductivity in bulk infinite-layer nickelate” by Kejun Hu et al.

Response to reviewers

We would like to express our gratitude once again to the reviewers for their continued support on our work. The revised manuscript has been crafted, incorporating all the comments and suggestions provided by the reviewers. We look forward to continued support from the reviewers.

For your convenience, detailed point-to-point responses to all raised comments are presented below. All page numbers, references, and figures correspond to the revised version of the manuscript. Changes have been indicated in red throughout the revised manuscript.

Sincerely yours

Dongsheng Song on behalf of all of the authors

Institutes of Physical Science and Information Technology

Anhui University, Hefei 230601, China

E-mail: dsong@ahu.edu.cn

Responses to the comments raised by the referees

Reviewer #1:

In their revised manuscript, Kejun Hu et al. have introduced a host of text additions and changes together with many new microscopy figures, in response to the comments of all three reviewer. I find that these changes have improved the manuscript significantly, and congratulate them for their efforts. The comments and queries from the reviewers have been treated seriously and in depth and, I believe, are all successfully addressed. The readability and messaging are also now much improved. Therefore, I do not hesitate to recommend the publication of this work, which will make a nice article for Nature Communications. Nevertheless, to maximize the standard of the work, I do have a few minor points which should first be addressed, as now detailed.

Response:

We thank the reviewer once again for his/her continued support in the publication of our work and greatly appreciate his/her thorough scrutiny of our paper. The revisions have been made point by point as detailed below to further improve the manuscript.

Line 100-101, proposed edit: "While the parent phase Nd_{0.8}Sr_{0.2}NiO₃ has an orthorhombic structure with (Pbnm) lattice parameters of $a_o = 5.39 \text{ \AA}$, $b_o = 5.38 \text{ \AA}$, and $c_o = 7.61 \text{ \AA}$, here we use the pseudocubic unit cell..."

Revision:

The sentence "The parent phase Nd_{0.8}Sr_{0.2}NiO₃ has an orthorhombic structure with lattice parameters of $a_o = 5.39 \text{ \AA}$, $b_o = 5.38 \text{ \AA}$, and $c_o = 7.61 \text{ \AA}$, the pseudocubic unit cell with lattice parameters of $a_{pc} = 3.808 \text{ \AA}$, $c = 3.805 \text{ \AA}$ to define the orientation."

has been revised as,

"While the parent phase Nd_{0.8}Sr_{0.2}NiO₃ has an orthorhombic structure (*Pbnm*) with lattice parameters of $a_o = 5.39 \text{ \AA}$, $b_o = 5.38 \text{ \AA}$, and $c_o = 7.61 \text{ \AA}$, here we use the pseudocubic unit cell to define the orientation with lattice parameters of $a_{pc} = b_{pc} = 3.808 \text{ \AA}$, $c_{pc} = 3.805 \text{ \AA}$."

Line 103, proposed edit: "... orientation relationship between..."

Revision:

The sentence "The orientation between the pseudocubic indices and the orthorhombic indices of the Nd_{0.8}Sr_{0.2}NiO₃ are $(010)_{pc} // (110)_O$, $(100)_{pc} // (110)_O$, $(001)_{pc} // (001)_O$."

has been revised as,

"The orientation relationship between the pseudocubic indices and the orthorhombic indices of the Nd_{0.8}Sr_{0.2}NiO₃ are $[010]_{pc} // [110]_O$, $[100]_{pc} // [1-10]_O$, $[001]_{pc} // [001]_O$."

Line 104: would be more systematic to have $(100)_{pc} // (1-10)_O$

In my opinion, it would also be preferable to define the orientation relationships by lattice vectors not planes (i.e. swap the round brackets for square ones).

Revision:

The sentence "The orientation between the pseudocubic indices and the orthorhombic indices of the Nd_{0.8}Sr_{0.2}NiO₃ are $(010)_{pc} // (110)_O$, $(100)_{pc} // (110)_O$, $(001)_{pc} // (001)_O$."

has been revised as,

“The orientation relationship between the pseudocubic indices and the orthorhombic indices of the $\text{Nd}_{0.8}\text{Sr}_{0.2}\text{NiO}_3$ are $[010]_{\text{pc}} // [110]_{\text{O}}$, $[100]_{\text{pc}} // [1-10]_{\text{O}}$, $[001]_{\text{pc}} // [001]_{\text{O}}$.”

Line 110: I do not find this line clear. If resistance is elevated by grain boundaries and voids, the material should become more insulating. From the phrase "deteriorate the insulating properties", it is not clear if the authors mean that it becomes more or less insulating...

Response:

Thanks for the reviewer’s comment. We apologize for the confusion caused by the wording in the manuscript. The presence of grain boundaries and voids increases the resistance and makes the material more insulating.

Revision:

The sentence “The presence of grain boundaries and voids in bulk nickelate, as shown in Fig. 1b, can indeed elevate resistance and deteriorate the insulating properties of the material.”

has been revised as,

“The presence of grain boundaries and voids in bulk nickelate, as shown in Fig. 1b, can indeed elevate resistance and make the material more insulating.”

Line 113: similarly, perhaps it should be "despite the reduction in insulating properties"?

Revision:

The sentence “However, despite the improvement in insulating properties under high-pressure conditions, superconducting properties remained elusive.”

has been revised as,

“However, despite the weakening of insulating behavior under high-pressure conditions, superconducting properties remained elusive.”

Line 115: more appropriate might be "... grain boundaries and voids is not the primary detrimental effect..."?

Revision:

The sentence “This suggests that the presence of grain boundaries and voids does not have a detrimental effect on superconductivity.”

has been revised as,

“This suggests that the presence of grain boundaries and voids is not the primary detrimental effect on superconductivity.”

Line 122: "... examination of the grain interiors..."

Revision:

The sentence “Further examination of the interior grains provides clear stripe contrast in Fig. 1e and 1f.”

has been revised as,

“Further examination of the grain interiors provides clear stripe contrast, as shown in Fig. 1e and 1f.”

Line 141-142: as above, "increase resistance and degrade the insulating properties" is confusing.

Revision:

The sentence “Consequently, the polycrystalline nature, in conjunction with the abundance of grain boundaries, is expected to increase resistance and degrade the insulating properties of the material, in contrast to single crystal thin films.”

has been revised as,

“Consequently, the polycrystalline nature, in conjunction with the abundance of grain boundaries, is expected to increase resistance and make the material more insulating, in contrast to single crystal thin films.”

Fig. 1(a). Would be more obviously consistent with the right-handed rule – and more consistent with panels (e) and (f) – if [010]_{pc} points to the right. Or, the left-pointing arrow should be indexed as [0-10]_{pc}.

Response:

Thanks for the reviewer’s comment, we have changed the direction of the arrows to make it consistent with the right-hand rule.

Revision:

Figure 1 has been updated in the revised manuscript as shown below.

Fig. 1 | Polycrystalline Nd_{0.8}Sr_{0.2}NiO₂ imaged by HAADF-STEM. a Atomic structure model with direction vectors of perovskite RNiO₃ and infinite layer RNiO₂ (R: Nd/Sr). **b** EDS mappings of Nd_{0.8}Sr_{0.2}NiO₂ at low magnification. **c** Temperature dependence of resistivity for bulk Nd_{0.8}Sr_{0.2}NiO₂ in the range of 2-300 K. **d** High-resolution HAADF-STEM image of Nd_{0.8}Sr_{0.2}NiO₂. **e** and **f** The magnified images of marked regions in **d**, with the corresponding SAED pattern in the insets.

Line 160: perhaps better as "the rock salt layers of the RP phase"?

Revision:

The sentence “Furthermore, the rock salt layers undergo a transformation into the fluorite structure after topotactic reduction, involving the rearrangement of oxygen atoms between two R layers, as illustrated by the red spheres in Fig. 2a.”

has been revised as,

“Furthermore, the rock salt layers of the RP phase undergo a transformation into the fluorite structure after topotactic reduction, involving the rearrangement of oxygen atoms between two *R* layers, as illustrated by the red spheres in Fig. 2a.”

Line 296-297: perhaps better as "Remarkably, there are still randomly-distributed residual apical oxygen atoms in the Nd/Sr plane..."?"

Revision:

The sentence “Remarkably, there is still a random distribution of residual apical oxygen in Nd/Sr plane after topotactic reduction, as indicated by the orange arrows in Fig. 5c - 5e.” has been revised as,

“Remarkably, there are still randomly distributed residual apical oxygen atoms in Nd/Sr planes after topotactic reduction, as indicated by the orange arrows in Fig. 5c - 5e.”

Line 356: the "while" seems unnecessary.

Response:

We have deleted “while” in line 356 of the revised manuscript.

Line 366: "Also, it's important to note..." -> "It is important to note..."

Revision:

The sentence “Also, it's important to note that many thin films, i.e. nickelates and Sr₂RuO₄, exhibit superconductivity even in the presence of a high density of defects similar to those found in bulk materials.”

has been revised as,

“It is important to note that many thin films, i.e. nickelates and Sr₂RuO₄, exhibit superconductivity even in the presence of a high density of defects similar to those found in bulk materials.”

Line 382: perhaps better as "This makes the NiO₂ planes buckle..."?"

Revision:

The sentence “This makes the buckling of the NiO₂ planes, leading to enhanced scattering of electrons and inhibiting the pairing.”

has been revised as,

“This makes the NiO₂ planes buckle, leading to enhanced scattering of electrons and inhibiting the pairing.”

Supp. Fig. S2(c): indexing inconsistent with right-hand rule.

Revision:

Supplementary Fig. S2 has been updated in the revised manuscript as shown below.

Supplementary Fig. S2 | Analysis of the interleaved regions by Scanning diffraction measurements based on the four-dimensional STEM (4D-STEM) technique. **a** and **c** The typical nano-beam electron diffraction patterns extracted from the two orthogonal regions in the reduced grain. **b** and **d** Virtual dark-field images reconstructed by selecting the 011 spots, respectively.

*Supp. Fig. S13: indexing of directions in yellow regions inconsistent with right-hand rule. Note [001] must be coming towards the viewer, because [100] is coming towards the viewer in the red regions. Either point [100] in the opposite direction, or label as [-100]. (*Or* do likewise for the [010] arrow.)*

Revision:

Supplementary Fig. S13 (Supplementary Fig.14 in the revised version) has been updated in the revised manuscript as shown below.

Supplementary Fig. S14 | Discontinuity of NiO₂ planes within the grains. **a** and **b** The red and yellow dotted boxes correspond to domains with the orientations of [100] and [001], respectively.

Reviewer #2:

I appreciate the efforts by Hu, et al. to improve the introduction and discussion of their manuscript, and the additional experiments which are now included. I feel that the topic of different kinds of disorder which are characterized here are useful information to the community, but for this same reason feel it will be crucial that their presented results are as accurate and accessible as possible, especially to a broad audience not familiar with microscopy techniques. I therefore have a few additional suggestions which I feel will greatly improve the clarity of the article for publication in Nature Communications.

Response:

We thank the reviewer once again for his/her continued support in the publication of our work and greatly appreciate his/her comments to make our paper more accurate and accessible. The revisions have been made point by point as detailed below to further improve the manuscript.

For the strengths of the article, I feel the title is not appropriate. It suggests a definitive or smoking gun solution to the “absence of superconductivity”, while the results here simply offer insight to the many complicated effects which are likely all contributing in some way. I suggest something along the lines of “Atomic scale disorder and reconstruction in bulk infinite-layer nickelates” would be more appropriate.

Response:

Thanks for the reviewer’s comment and helpful suggestion. We agree with the reviewer that the original title was not definitive enough and have revised it as “**Atomic scale disorder and reconstruction in bulk infinite-layer nickelates lacking superconductivity**”.

I do not understand the argument in lines 113: “despite the improvement in insulating properties under high-pressure conditions, superconducting properties remained elusive. This suggests that the presence of grain boundaries and voids does not have a detrimental effect on superconductivity”. Are the authors trying to argue that the intra-grain crystalline regions are themselves insulating? This is confusing to me, since other reports of bulk square-planar nickelate single crystals exhibit metallic transport (though still no superconductivity), see Li et al. Sci. Adv. 9, eade4418 (2023) 10.1126/sciadv.ade4418 or <https://arxiv.org/pdf/2403.00493.pdf>. If the authors argue that the voids and boundaries do not significantly influence the electronic transport, can they include a resistivity trace for the precursor powder? Presumably a high-quality precursor should also exhibit metallic behavior until at least 100-200 K (even lower if properly doped).

Response:

Thanks for the reviewer’s comments and providing these valuable references. We apologize for the confusing statements regarding the insulating behaviors of the reduced bulk material. As pointed out by the reviewer, we would like to argue that the presence of voids and boundaries is not the determining factor for the absence of superconductivity. The presence of grain boundaries and voids mostly increases the resistance and makes the material more insulating. Following the reviewer’s suggestion, we have included the temperature-dependent resistance data for the precursor $\text{Nd}_{0.8}\text{Sr}_{0.2}\text{NiO}_3$ in Supplementary Fig. 12b as shown below, which shows metallic properties above 10 K as pointed out by the reviewer.

With regard to “*other reports of bulk square-planar nickelate single crystals exhibit metallic transport*”, we believe this phenomenon may be related to different rare earth elements and structural variations. For example, $\text{La}_4\text{Ni}_3\text{O}_8$ [*Nat. Phys.* **13**, 864-869, 2017] and $\text{Nd}_4\text{Ni}_3\text{O}_8$ [*J. Phys.* :

Condens. Matter **33**, 075503, 2020] display the insulating states, while $\text{Pr}_4\text{Ni}_3\text{O}_8$ [*Sci. Adv.* **9**, eade4418, 2023] is metallic. $\text{La}_{0.93}\text{Ca}_{0.07}\text{NiO}_3$ [*Sci. Adv.* **7**, eabl8091, 2021] is metallic, while NdNiO_3 is strongly insulating at low temperatures [*Nature* **572**, 624-627, 2019].

Revision:

The misleading sentence “However, despite the improvement in insulating properties under high-pressure conditions, superconducting properties remained elusive. This suggests that the presence of grain boundaries and voids does not have a detrimental effect on superconductivity.”

has been revised as

“However, despite the weakening of insulating behavior under high-pressure conditions, superconducting properties remained elusive. This suggests that the presence of grain boundaries and voids is not the primary detrimental effect on superconductivity.”

The misleading sentence “The presence of grain boundaries and voids in bulk nickelate, as shown in Fig. 1b, can indeed elevate resistance and deteriorate the insulating properties of the material.”

has been revised as,

“The presence of grain boundaries and voids in bulk nickelate, as shown in Fig. 1b, can indeed elevate resistance and make the material more insulating.”

The R-T data for the precursor $\text{Nd}_{0.8}\text{Sr}_{0.2}\text{NiO}_3$ powers has been involved in **Supplementary Fig. S12** in the revised manuscript as shown below.

Supplementary Fig. S12 | Structure characterization of bulk $\text{Nd}_{0.8}\text{Sr}_{0.2}\text{NiO}_3$ before topotactic reduction. **a** EDS mapping of $\text{Nd}_{0.8}\text{Sr}_{0.2}\text{NiO}_3$ polycrystalline at low magnification. **b** Temperature dependence of resistivity for bulk $\text{Nd}_{0.8}\text{Sr}_{0.2}\text{NiO}_3$ in the range of 2-300 K. **c** and **e** High-resolution BF-STEM image of $\text{Nd}_{0.8}\text{Sr}_{0.2}\text{NiO}_3$. **d** and **f** The magnified HAADF-STEM images of marked regions in **c** and **e**.

The EDX maps presented in Figure 3 appear highly aliased. My guess would be an aggressive Gaussian or Weiner filter applied by the EDX acquisition and processing software. I suggest the authors replace what is currently presented with the unfiltered data, or carefully choose a much lower filter and report it in the methods with unfiltered data in the Supplement. Likely also in Figure

1 and Supplemental Figure S11.

Response:

Thanks for the reviewer's comments and suggestions. Regarding the EDS mappings in Fig. 1 and Supplemental Fig. S11 (Supplemental Fig. S12 in the revised version) at low magnification, we have updated these figures with the unfiltered data. For the high magnification EDS mappings in Fig. 3, we have utilized a post-filter setting (average pixel size: 7, which is 15 in our previous manuscript), and the details of this are described in the Methods part. The unfiltered data for Fig. 3 is also shown in Supplemental Fig. S6.

Revision:

Figure 1 and Supplementary Fig. S12 have been updated in the revised manuscript as shown below.

Fig. 1 | Polycrystalline $\text{Nd}_{0.8}\text{Sr}_{0.2}\text{NiO}_2$ imaged by HAADF-STEM. a Atomic structure model with direction vectors of perovskite RNiO_3 and infinite layer RNiO_2 (R : Nd/Sr). **b** EDS mappings of $\text{Nd}_{0.8}\text{Sr}_{0.2}\text{NiO}_2$ at low magnification. **c** Temperature dependence of resistivity for bulk $\text{Nd}_{0.8}\text{Sr}_{0.2}\text{NiO}_2$ in the range of 2-300 K. **d** High-resolution HAADF-STEM image of $\text{Nd}_{0.8}\text{Sr}_{0.2}\text{NiO}_2$. **e** and **f** The magnified images of marked regions in **d**, with the corresponding SAED pattern in the insets.

Supplementary Fig. S12 | Structure characterization of bulk $\text{Nd}_{0.8}\text{Sr}_{0.2}\text{NiO}_3$ before topotactic reduction. **a** EDS mapping of $\text{Nd}_{0.8}\text{Sr}_{0.2}\text{NiO}_3$ polycrystalline at low magnification. **b** Temperature dependence of resistivity for bulk $\text{Nd}_{0.8}\text{Sr}_{0.2}\text{NiO}_3$ in the range of 2-300 K. **c** and **e** High-resolution BF-STEM image of $\text{Nd}_{0.8}\text{Sr}_{0.2}\text{NiO}_3$. **d** and **f** The magnified HAADF-STEM images of marked regions in **c** and **e**.

The following statements have been added to the revised manuscript,
Page 10, Line 191,

“For the high-magnification EDS mapping in Fig. 3a, a lower filter setting is utilized, as described in the **Methods** section. The unfiltered data for Fig. 3a was also shown in Supplementary Fig. S6.”

Page 22, Line 451, Methods section,

“Regarding the EDS mappings in Fig. 1 and Supplemental Fig. S12 at low magnification, we have utilized the unfiltered data. For the high magnification EDS mappings in Fig. 3, we have utilized a post-filter setting (average pixel size: 7) to smooth these images.”

Figure 3 has been updated in the revised manuscript as shown below.

Fig. 3 | EDS and EELS measurements of $\text{Nd}_{0.8}\text{Sr}_{0.2}\text{NiO}_2$. **a** EDS elemental mappings of the stripe

structure in $\text{Nd}_{0.8}\text{Sr}_{0.2}\text{NiO}_2$. **b** The intensity profile of elemental distributions in **a**. **c** EELS O *K*, Ni *L*, and Nd *M* of infinite-layer phase (black) and T'-type phase (red).

Supplementary Fig. S6 has been updated in the revised manuscript as shown below.

Supplementary Fig. S6 | EDS and EELS measurements. **a** Unfiltered EDS elemental mappings in **Fig. 3a**. **b** EELS acquisition at different locations in **Fig. 3c**. To improve the signal-to-noise ratio, integrate the signal over the regions as indicated by the red boxes. **c** EELS mapping at atomic resolution of the T'-type phases.

In figure 5, the color bars in panels g and i are confusing because they present continuum parameters on a diverging colormap. The figures would carry much more physical intuition if the authors remap the red-blue color scale so that the middle value (white) corresponds to a physically relevant “middle” value, eg. the average c/a ratio of 0.86 (or 0.886) and 180 degree bond angle.

Response:

Thanks for the reviewer’s helpful suggestion. We have remapped the red-blue color scale in Fig. 5g and 5i so that the middle value (white) corresponds to a physically relevant “middle” value, specifically the average c/a ratio of 0.886 and the bond angle of 180° . Additionally, a divergent colormap has been employed in the updated Fig. 5.

Revision:

Figure 5 has been updated in the revised manuscript as shown below.

Fig. 5 | Atomic iDPC-STEM imaging of infinite-layer phase. **a** Structure transition during the topotactic reduction. **b** iDPC-STEM image of infinite-layer phase at [100] orientation. **c**, **d** and **e** Enlarged iDPC-STEM images extracted from the orange dotted line box in **b**, where the orange arrow marks the residual oxygen atom in the Nd/Sr atomic plane. **f** Atom model of the distorted NiO₂ layers projected in [100] direction. The bond angle and the deviation of Ni atom are marked as well. **g** The ratio of lattice parameter c/a according to **b**. **h** Statistical data of lattice constants a and c . **i** The calculated bond angle contour maps are plotted over the iDPC-STEM image. **j** The histogram displays the statistical data for bond angles measured across multiple regions, with a fitted value of $177.0^\circ \pm 5.3^\circ$.

Actually, I do not find that the bond angle mapping by iDPC adds much useful information at all to the rest of the manuscript, beyond what one could already expect from the many other kinds of disorder discussed. Furthermore, given that the Ni displacements shown in Figure 5i bond angle map all show a systematic “up” component and given the extreme sensitivity of iDPC to very small sample mistilt, thickness, and other projection artifacts, I wonder about the robustness of these results. In regard to the speculation that “it could therefore be inferred that the strain could suppress the local distortions and potentially flatten the NiO₂ planes”, I do not understand why intuitively one would expect a square planar phase to have any kind of systematic bond distortion along one direction, which would impart the bulk compound with an overall mirror symmetry-breaking. To me, this part of the discussion feels far too speculative, and without significantly more effort to demonstrate its reproducibility I would be hesitant to include it here.

Response:

Thanks for the reviewer's insightful comments and suggestions. We also extend our respect for the reviewer's expertise in electron microscopy. In response to the reviewer's hesitation about including the discussion of bond angles and Ni displacements in the main text, we have removed the results of Ni displacements in the revised manuscript. Additionally, we have made some efforts to map the

O-Ni-O bond angles across various regions in multiple experiments to ensure the reproducibility and reliability of our results, as shown in Supplementary Fig. S11a below. The average bond angle deviates from 180° with a value of $177.0^\circ \pm 5.3^\circ$. For comparative purposes, we also analyzed cubic SrTiO_3 , where the measured bond angle is $180.3^\circ \pm 2.6^\circ$, as shown in Supplementary Figs. S11b. Although it is difficult to maintain absolutely identical imaging conditions, e.g. aberrations, sample orientation and thickness, this comparison lends some validity to our measurements on $\text{Nd}_{0.8}\text{Sr}_{0.2}\text{NiO}_2$. Therefore, we seek the reviewer's opinion on including the information of the bond angle here. Regarding that *"it could therefore be inferred that the strain could suppress the local distortions and potentially flatten the NiO_2 planes"*, we have removed these speculative statements in the revised manuscript as suggested by the reviewer.

Revision:

The paragraph "The displacement of Ni atoms is plotted here in Fig. 5i with yellow arrows, and the contour map of the bond angle is overlaid together. Obviously, the bond angles O-Ni-O are largely deviated from the 180° , thereby suppressing the flatness of superconducting NiO_2 layers. The regions with large distortions of the NiO_2 plane correspond to a large displacement of Ni as indicated by arrows in Fig. 5i. The disorders within the NiO_2 layers are consistent with previous X-ray analysis of bulk polycrystalline samples with broaden diffraction peaks³⁶. These observations suggest the complicated distortions in Ni-O planes for polycrystalline $\text{Nd}_{0.8}\text{Sr}_{0.2}\text{NiO}_2$ after chemical reduction. Referring to the widely observed superconducting characters in nickelate thin films, it could therefore be inferred that the strain could suppress the local distortions and potentially flatten the NiO_2 planes, which might be necessary for accommodating the superconductivity but absent in its bulk form⁵¹⁻⁵⁴."

has been revised as,

"The contour map of the O-Ni-O bond angle is displayed in Fig. 5i, where it is noticeably deviated from 180° , potentially impacting the flatness of the NiO_2 layers. To ensure the reproducibility and reliability of our measurements, we have calculated the O-Ni-O bond angles across various regions in multiple experiments, as detailed in Supplementary Fig. S11. The average bond angle is determined to be $177.0^\circ \pm 5.3^\circ$, as shown in Fig. 5j, with a broadened distribution. This finding highlights the presence of complex distortions in the Ni-O planes of polycrystalline $\text{Nd}_{0.8}\text{Sr}_{0.2}\text{NiO}_2$ after chemical reduction. These disorders within the NiO_2 layers are consistent with previous X-ray analysis of bulk polycrystalline samples, which have shown broad diffraction peaks³⁶."

Figure 5 has been updated in the revised manuscript as shown below.

Fig. 5 | Atomic iDPC-STEM imaging of infinite-layer phase. **a** Structure transition during the topotactic reduction. **b** iDPC-STEM image of infinite-layer phase at [100] orientation. **c**, **d** and **e** Enlarged iDPC-STEM images extracted from the orange dotted line box in **b**, where the orange arrow marks the residual oxygen atom in the Nd/Sr atomic plane. **f** Atom model of the distorted NiO₂ layers projected in [100] direction. The bond angle and the deviation of Ni atom are marked as well. **g** The ratio of lattice parameter c/a according to **b**. **h** Statistical data of lattice constants a and c . **i** The calculated bond angle contour maps are plotted over the iDPC-STEM image. **j** The histogram displays the statistical data for bond angles measured across multiple regions, with a fitted value of $177.0^\circ \pm 5.3^\circ$.

Supplementary Fig. S11 has been added to the revised manuscript as shown below.

Supplementary Fig. S11 | The O-Ni-O bond angles in bulk Nd_{0.8}Sr_{0.2}NiO₂. **a** The frequency histogram of the bond angle for bulk Nd_{0.8}Sr_{0.2}NiO₂ measured across various regions in multiple experiments. The average bond angle deviates from 180° with a value of $177.0^\circ \pm 5.3^\circ$. **b** The frequency histogram of the bond angle for cubic SrTiO₃. For comparative purposes, we analyzed cubic SrTiO₃ here, where the measured bond angle is $180.3^\circ \pm 2.6^\circ$. Although it is difficult to

maintain absolutely identical imaging conditions, e.g. aberrations, sample orientation and thickness, this comparison lends some validity to our measurements on $\text{Nd}_{0.8}\text{Sr}_{0.2}\text{NiO}_2$.

Minor comments:

Lines 69-70 “Yet, only a specific case, $\text{Nd}_6\text{Ni}_5\text{O}_{12}$ with $n = 5$, has been found to exhibit superconductivity with outer electrons $d_{8.8}$ ” – I suggest removing this comment, as results recently presented by the Mundy group suggest other layered phases can also host superconductivity (<https://meetings.aps.org/Meeting/MAR24/Session/G16.5>).

Response:

Thanks for the reviewer's information and suggestion. We have removed this sentence in the revised manuscript.

The legend reference in supplementary figure 7 to “Berit H. Goodge” seems like a bit of an odd way to reference published work – perhaps the authors can replace it with a simple citation number or something like “Goodge, et al.”

Response:

Thanks for the reviewer's suggestion. We have used “Goodge, et al.” in the legend of Supplementary Fig. S7.

Revision:

Supplementary Fig. S7 | The EELS data of $\text{Nd}_{0.8}\text{Sr}_{0.2}\text{NiO}_2$ in this manuscript compared with those of Goodge, et al.³¹. The weak pre-peak of O K edge corresponds to the underdoped level of Sr, as indicated by the red arrows. The shift of Ni $L_{3,2}$ edges towards lower energy is consistent with the lower Sr doping, despite the broadening of the peaks, which might be attributed to the relatively lower EELS energy resolution in our experiments (~ 1 eV).

Reviewer #3:

The authors of the manuscript have done a fairly good job responding to the previous reviews. The manuscript is even better than the initial submission, which was also good. The new supporting figures are very nice. However, they did miss a couple of the key comments addressed in the first review. I think that as a whole the authors conclusions are correct, however there is a possibility of miss interpreted EELS data that can only be addressed with more robust data analysis and thought into physical origins, and not simple rewording of sentences. I suggest that the EELS be given more thought before publication. However, given that the different characterization techniques lead to the same conclusion and the conclusions are logical. I feel the manuscript if left as would still be of high quality and leave it to the editor's decision if the suggested EELS consideration are necessary.

Response:

We thank the reviewer once again for his/her continued support in the publication of our work and greatly appreciate his/her evaluation of our new data. We also apologize for the missing revisions in our previous manuscript. The revisions have been made point by point as detailed below to further improve the manuscript.

More details:

Question 1: They included a new supporting figure where the streaks in the diffraction pattern are clearer, but they have not changed the main text figure as they stated in the response. The streaks in the diffraction patterns in Fig1 d and e are still not visible but are main discussion topics of the text.

Response:

We thank the reviewer for his/her pertinent comment. We apologize for the error in making the revision in the previous version. We have now modified the diffraction patterns in Fig. 1e and 1f to make them clearer.

Revision:

Figure 1 has been updated in the revised manuscript as shown below.

Fig. 1 | Polycrystalline $\text{Nd}_{0.8}\text{Sr}_{0.2}\text{NiO}_2$ imaged by HAADF-STEM. **a** Atomic structure model with direction vectors of perovskite RNiO_3 and infinite layer RNiO_2 (R : Nd/Sr). **b** EDS mappings of $\text{Nd}_{0.8}\text{Sr}_{0.2}\text{NiO}_2$ at low magnification. **c** Temperature dependence of resistivity for bulk $\text{Nd}_{0.8}\text{Sr}_{0.2}\text{NiO}_2$ in the range of 2-300 K. **d** High-resolution HAADF-STEM image of $\text{Nd}_{0.8}\text{Sr}_{0.2}\text{NiO}_2$. **e** and **f** The magnified images of marked regions in **d**, with the corresponding SAED pattern in the insets.

Question 4:

The change could still be from correlation and not valence state change. It is in general difficult to understand the physical correlation to ELNES in such a complex system that has so many possible things effecting electronic fine structure. The Goodge, B. H. reference makes a point of this by showing that the Ni-L3 edge is sensitive to doping and oxygen content. The authors make a comment that the Sr-doping decreases the Ni-L3 edge energy, but according to the Goodge, B. H. reference the Ni-L3 edge should increase with Sr and is also sensitive to the oxygen content. The authors have mentioned that the valence state of Ni has changed based on the intensity but do not see any peaks shifts. In addition, the cited paper shows that the Ni-L3 peak energy and shape is sensitive to the oxygen content, while the Ni-L2 edge shape changes and the onset remains the same energy. Again, the authors see a change in the intensity ratio but no change in the peak energies, which is odd and could suggest that something else is resulting in fine structure changes. Although valence state change is most likely the cause for the different intensity ratio's other things could be happening and the other changes observed with valence state changes in other studies are not observed in this manuscript. The authors adjusted their wording accommodate my comment regarding changes in molecular orbital theory and crystal field splitting but did not dive into the other physics that might be resulting in fine structures changes.

Response:

We appreciate the reviewer's detailed comments and extend our respect for the reviewer's expertise in EELS analysis. We thank the reviewer for providing this information and fully agree with the reviewer that establishing a physical correlation to ELNES in such a complex system is challenging. Factors such as Sr doping, variations in oxygen content, and changes in the coordination

environment can alter the shape, intensity, and energy position of the $L_{3,2}$ edges for Ni. Therefore, it is not entirely rigorous to simply attribute the large L_3/L_2 ratio to a valence change of Ni in our previous manuscript.

Given that the Ni L_3 peak energy is expected to increase with higher Sr concentrations and is also sensitive to the oxygen content, as discussed by Goodge et al. and succinctly summarized by the reviewer, the decrease in the Ni L_3 peak energy observed in our experiments could be attributed not only to the lower doping level of Sr in the infinite layers, but also to variations in oxygen content, due to the presence of residual apical oxygen atoms. However, the exact physical correlation is not yet fully understood to our knowledge at present.

For the comparison of Ni $L_{3,2}$ edges between the T'-phase and the infinite layer, we observed a large ratio of L_3/L_2 for the T'-type phase, while the peak energies of L_3 and L_2 edges remained nearly unchanged between these two phases. As the reviewer pointed out, it is odd to observe a change in the intensity ratio without corresponding changes in peak energies. Therefore, we believe that in addition to potential valence changes, the different coordination environments of Ni between the T'-phase and the infinite layer might contribute to this observed phenomenon.

Although the direct physical correlation to ELNES cannot be easily and definitely addressed here due to the complexity of this system, we would like to include the aforementioned discussions to give the possible reasons for our observations.

Revision:

We have changed the following statements in the revised manuscript,

The paragraph “Although the fine structures of the Nd M edge show no difference, the prominent increased intensity of the L_3 edge for Ni is observed for T'-type phase. The larger ratio of L_3/L_2 corresponds to a higher valence of Ni, which is attributed to the alterations in crystal field splitting arising from changes in the coordination environment of Ni. The resultant valence state of Ni leads to the modified $3d$ -orbital electron filling and thus non-superconducting states according to the superconducting phase diagram^{30, 43}. The segregation of Sr near fluorite layers may result in lower doping within the infinite-layer regions, potentially responsible for the absence of superconductivity. To explore this possibility, we compare the EELS results of the infinite-layer phases with those in superconducting infinite-layer nickelates thin films, which are nicely and systematically measured in a prior study³¹. As shown in Supplementary Fig. S7, the pre-peak of the O K edge, indicative of the doping level³¹, is weaker than that in the superconducting thin films. This observation aligns with the underdoped concentration of Sr in the infinite-layer phase due to Sr segregation near fluorite layers. The shift of Ni $L_{3,2}$ edges towards lower energy is also consistent with the lower Sr doping, despite the broadening of the peaks, which might be attributed to the relatively lower EELS energy resolution in our experiments (~ 1 eV). As the infinite-layer structure and the proper concentration of doped Sr are essential to superconductivity, the presence of various T'-type phases with such imperfections would probably lead to the insulating and non-superconducting characters in nickelates^{20,36,44}.”

has been revised as,

“Although the fine structures of the Nd M edge show no difference, the prominent increased intensity of the L_3 edge for Ni is observed for T'-type phase. The larger ratio of L_3/L_2 is generally attributed to the valence changes, i.e. a higher valence of Ni in the T'-type phase. However, the peak energies of L_3 and L_2 edges remained nearly unchanged between these two phases. In this complex

system, factors such as Sr doping, variations in oxygen content, and changes in the coordination environment can alter the shape, intensity, and energy position of the $L_{3,2}$ edges for Ni³¹. Therefore, we believe the different coordination environments of Ni between the T'-type phase and the infinite layer might contribute to this observed phenomenon.

The segregation of Sr near fluorite layers may result in lower doping within the infinite-layer regions, potentially responsible for the absence of superconductivity. To explore this possibility, we compare the EELS results of the infinite-layer phases with those in superconducting infinite-layer nickelates thin films, which are nicely and systematically measured in a prior study³¹. As shown in Supplementary Fig. S7, the pre-peak of the O K edge, indicative of the doping level³¹, is weaker than that in the superconducting thin films. This observation aligns with the underdoped concentration of Sr in the infinite-layer phase due to Sr segregation near fluorite layers. The shift of Ni $L_{3,2}$ edges towards lower energy in Supplementary Fig. S7b is also consistent with the lower Sr doping, despite the broadening of the peaks, which might be attributed to the relatively lower EELS energy resolution in our experiments (~ 1 eV). However, given that the Ni- L_3 peak energy is expected to increase with higher Sr concentrations and is also sensitive to the oxygen content³¹, the decrease in the Ni L_3 peak energy observed here could be attributed not only to the lower doping level of Sr in the infinite layers, but also to variations in oxygen content, due to the presence of residual apical oxygen atoms. As the infinite-layer structure, the proper concentration of doped Sr and the oxygen content are essential to superconductivity, the presence of various T'-type phases with such imperfections would probably lead to the insulating and non-superconducting characters in nickelates^{20,36,44}.”

Question 6:

It sounds like the authors were careful to manage the current during spectroscopy and evaluate possible beam damage. However, they have presented none of the evidence and this is a question that will be asked by any expert in the field who read the paper. The current is not the main issue when it comes to damage. The main factors are accumulated dose and operating energy. 300 kV is an energy that most oxides damage at, especially with the dose requirements to perform ELNES analysis as performed in the manuscript. I know that this was a consideration for Goodge, B. H. whom the authors cite. I believe that the authors have been thorough in their experiments, but I ask that the authors add in data regarding exposure to the beam.

Response:

Thanks for the reviewer's comment. Following the reviewer's suggestion and the methodology detailed in Goodge, B. H.'s paper, we have added data for the EELS O K and Ni L edges of Nd_{0.8}Sr_{0.2}NiO₂ under the accumulated dose in Supplementary Fig. S16. The experimental parameters were consistent with those used for our EELS measurements, corresponding to an electron dose of $\sim 4 \times 10^6$ e⁻/Å² for each STEM-EELS mapping. We conducted EELS mappings over the same region eight times. The averaged EELS signals are shown in Supplementary Fig. S16 below. No significant modification of the O K and Ni L edges is observed.

Revision:

In the Methods section, we have changed the following statements,

The sentence “The O- K edge, which is sensitive to the irradiation damage in oxides, was also examined before and after STEM-EELS acquisition, with no noticeable changes observed.”

has been revised as,

“To evaluate the effect of electron dose on sample damage, we examined the O *K* edge and Ni *L* edge, which are sensitive to irradiation damage in oxides, during STEM-EELS acquisition. We conducted EELS mappings over the same region eight times under identical experimental conditions, corresponding to an electron dose of approximately $4 \times 10^6 \text{ e}^-/\text{\AA}^2$ for each measurement. The averaged EELS signals are displayed in Supplementary Fig. S16. No significant modification of the O *K* and Ni *L* edges was observed.”

Supplementary Fig. S16 has been added to the revised manuscript as shown below.

Supplementary Fig. S16 | Effect of electron-beam on the EELS O *K* and Ni *L* edges of $\text{Nd}_{0.8}\text{Sr}_{0.2}\text{NiO}_2$ under accumulated dose. The experimental parameters were consistent with those used for our EELS measurements in the main text, corresponding to an electron dose of $\sim 4 \times 10^6 \text{ e}^-/\text{\AA}^2$ for each STEM-EELS mapping. The EELS mappings were conducted over the same region eight times. The averaged EELS signals are displayed. No significant modification of the O *K* and Ni *L* edges is observed.

REVIEWERS' COMMENTS

Reviewer #2 (Remarks to the Author):

I appreciate the authors' careful consideration and response to my previous suggestions. With regards to their analysis of Ni-O bond angles, I appreciate the addition of benchmarking to SrTiO₃ and statistical analysis over many experiments, and find their updated discussion of the data and its potential limitations very helpful as currently presented.

I find the manuscript in its current form to be both thorough and of great use to the nickelate community, and recommend its publication.

Reviewer #3 (Remarks to the Author):

In the revised manuscript, Kejun Hu et al. have addressed many of the reviewer concerns. Specific to my comments, the new figures provide the information requested clearly and the ELNES discussion now explores all aspects that might contribute to the observed changes. The changes have improved the manuscript and I see that it is fit for publication.